

# Cas9/gRNA-mediated genome editing of yeast mitochondria and *Chlamydomonas* chloroplasts

Byung-Chun Yoo, Narendra S. Yadav, Emil M. Orozco Jr and Hajime Sakai

NAPIGEN, Inc., Wilmington, DE, USA

## ABSTRACT

We present a new approach to edit both mitochondrial and chloroplast genomes. Organelles have been considered off-limits to CRISPR due to their impermeability to most RNA and DNA. This has prevented applications of Cas9/gRNA-mediated genome editing in organelles while the tool has been widely used for engineering of nuclear DNA in a number of organisms in the last several years. To overcome the hurdle, we designed a new approach to enable organelle genome editing. The plasmids, designated "Edit Plasmids," were constructed with two expression cassettes, one for the expression of Cas9, codon-optimized for each organelle, under promoters specific to each organelle, and the other cassette for the expression of guide RNAs under another set of promoters specific to each organelle. In addition, Edit Plasmids were designed to carry the donor DNA for integration between two double-strand break sites induced by Cas9/gRNAs. Each donor DNA was flanked by the regions homologous to both ends of the integration site that were short enough to minimize spontaneous recombination events. Furthermore, the donor DNA was so modified that it did not carry functional gRNA target sites, allowing the stability of the integrated DNA without being excised by further Cas9/gRNAs activity. Edit Plasmids were introduced into organelles through microprojectile transformation. We confirmed donor DNA insertion at the target sites facilitated by homologous recombination only in the presence of Cas9/gRNA activity in yeast mitochondria and *Chlamydomonas* chloroplasts. We also showed that Edit Plasmids persist and replicate in mitochondria autonomously for several dozens of generations in the presence of the wild-type genomes. Finally, we did not find insertions and/or deletions at one of the Cas9 cleavage sites in Chloroplasts, which are otherwise hallmarks of Cas9/gRNA-mediated non-homologous end joining (NHEJ) repair events in nuclear DNA. This is consistent with previous reports of the lack of NHEJ repair system in most bacteria, which are believed to be ancestors of organelles. This is the first demonstration of CRISPR-mediated genome editing in both mitochondria and chloroplasts in two distantly related organisms. The Edit Plasmid approach is expected to open the door to engineer organelle genomes of a wide range of organisms in a precise fashion.

Corresponding author
Hajime Sakai,
hajime.sakai@napigen.com

---

## INTRODUCTION

Organelles, such as mitochondria and chloroplasts, are cellular components of eukaryotic cells. They serve as sites of specific cellular functions: mitochondria are the sites for respiration, producing most of the cellular energy, ATP and NADH, while chloroplasts are primarily the sites for photosynthesis, converting light energy into cellular energy and sugar compounds in photosynthetic eukaryotes. Organelles also play an important role in metabolism, producing a number of key metabolites through their unique biochemical pathways such as tricarboxylic acid cycle, fatty acid β-oxidation and Calvin cycle.

For biotechnology applications, they also offer the potential as novel production platforms in eukaryotic cells. They are already sites for the biosynthesis of several valuable metabolites (e.g., amino acids, lipids, nucleic acids), including secondary metabolites related to nutraceutical and pharmaceutical compounds (*Almaraz-Delgado et al., 2014*; *Nielsen et al., 2016*). Furthermore, organelles provide efficiency in pathway engineering since the metabolic intermediates can be better channeled into desirable products without dilution with the larger pool of cytoplasmic metabolites (*Viitanen et al., 2004*; *Avalos, Fink & Stephanopoulos, 2013*; *Jin & Daniell, 2015*).

Mitochondria and chloroplasts have their own genomes. Organelle DNA provides a particular advantage because it can allow very high transgene expression through high copy number, as each cell can contain over 1,000 copies of organelle DNA (*Morley & Nielsen, 2016*). A typical plant leaf cell can contain up to 1,000–2,000 copies of the plastid DNA. This copy number advantage has already been exploited in plant chloroplasts, where a heterologous protein comprised up to 70% of leaf soluble cell protein in a plastid transgenic line in tobacco (*Oey et al., 2009*).

Because of the importance of organelles, several key technologies have been developed to allow manipulation of organelle DNA in the past. Examples include DNA transformation in yeast mitochondria (*Fox, Sanford & McMullin, 1988*), *Chlamydomonas* and higher plant chloroplasts (*Boynton et al., 1988*; *Svab, Hajdukiewicz & Maliga, 1990*), as well as high protein expression in tobacco chloroplasts (*McBride et al., 1995*). Manipulations of organelle DNA in the past have relied on the integration of engineered DNA into the organelle genomes through homologous recombination and subsequent selection of transgenic DNA by selectable markers, such as the *aadA* gene that confers resistance to spectinomycin.

While the activity of homologous recombination is relatively high in organelles, it is desirable to have a tool like CRISPR to allow precise genome manipulations. Such a tool will be useful for applications like gene therapy for treatment of human diseases caused by mitochondrial mutations (*Leslie, 2018*) and for engineering traits of crop plants such as biotic/abiotic stress tolerance, better $CO_2$ fixation and better yield by hybridization through altering organelle functions (*Ishiga et al., 2017*; *Estavillo et al., 2011*; *Cummins, Kannappan & Gready, 2018*; *Hanson & Bentolila, 2004*). Such applications in agriculture are widely considered non-GMO (*Jaganathan et al., 2018*). Nevertheless, unlike nuclear genomes, organelle genomes haven't received much attention from the recent applications of genome editing technologies due to the double membranes that

enclose the organelles and prevent the import of most nucleic acids (*Leslie, 2018*). Since CRISPR requires guide RNA and, in some cases, template DNA for donor DNA insertion at target sites in addition to Cas9-type endonuclease protein, such barriers are crucial obstacles for the application of the advanced genome editing approach in organelles. This would reflect why few successful attempts were reported with regard to CRISPR on organelle DNA in the past, while numerous successful applications on nuclear DNA have been published in the last several years. One such rare report describes a reduced amount of the mitochondrial gene that was targeted by simple overexpression of Cas9/guide RNA in the cytoplasm of human cells (*Jo et al., 2015*). However, it lacks any molecular evidence of cleavages induced by CRISPR to support their findings. In addition, donor DNA insertion at target sites will not be possible through such an approach because of the difficulty in importing template DNA into mitochondria.

To overcome the difficulty of organelle genome editing, we developed a new approach of organelle genome engineering for the deployment of the CRISPR system. Our approach utilizes plasmids ("Edit Plasmids") that are designed to replicate autonomously in organelles. A typical Edit Plasmid carries a cassette for the organelle-specific expression of Cas9-type endonuclease, a cassette for the expression of guide RNA(s), donor DNA, and a selectable or screenable marker. As a proof of concept, we designed and constructed prototypic versions of Edit Plasmids and introduced them into mitochondria of the yeast, *Saccharomyces cerevisiae*, and chloroplasts of the alga, *Chlamydomonas reinhardtii*, using microprojectile-based transformation. In both experimental systems, we found successful donor DNA insertion only when Cas9/gRNA were present in the Edit Plasmids. In addition, we analyzed DNA sequence at a gRNA target site in transgenic *C. reinhardtii* chloroplasts. We did not find Cas9-dependent SNPs or INDELs, which are usually hallmarks of nuclear genome editing through NHEJ DNA repair processes. As an evidence of autonomous replication, we were also able to rescue the Edit Plasmid from yeast cells that were grown for more than 20 generations after the cross between two strains carrying the Edit Plasmid and the wild-type mitochondrial genome. Our results demonstrate that the Edit Plasmids represent effective tools for organelle genome editing and writing overall.

## MATERIALS AND METHODS

### Cell cultures and chloroplast transformation in *C. reinhardtii*

The wild-type line of *C. reinhardtii* (CC-125) was obtained from the *Chlamydomonas* Resource Center (www.chlamycollection.org, University of Minnesota, St. Paul, MN, USA). Edit Plasmids were transformed into CC-125 using PDS-1000He particle delivery system (Bio-Rad) according to the methods described in *Barrera, Gimpel & Mayfield (2014)* and *Ramesh, Bingham & Webber (2011)*. Chloroplast transformants were selected on the Tris-Acetate-Phosphate (TAP) media supplemented with 100 µg/ml of spectinomycin under the light intensity of 50 µmol/m$^2$ s. The TAP media were made as described at the above-mentioned website of the *Chlamydomonas* Resource Center.

## Design of sgRNAs for *Chlamydomonas* chloroplast DNA

Guide RNA target sites were selected in the *psaA* gene (Gene ID: 2717000) encoded in the chloroplast genome of *C. reinhardtii* (NCBI Accession: NC_005353). The *psaA* gene encodes Photosystem I P700 chlorophyl a apoprotein A1, involved in photosynthesis and essential for photoautotrophic growth (https://www.uniprot.org/uniprot/P12154). To help design and select guide RNA target sites, a web-based Bioinformatics program, CRISPOR (http://crispor.tefor.net/; *Haeussler et al., 2016*) was employed. The gRNA targeting sequences chosen were all in exon 4 of the *psaA* gene: 5′GGTTTAAACCCT GTTACTGGTGG3′ (sgRNA1c), 5′CTTCACCTGTAAATGGACCACGG3′ (sgRNA2c), 5′TTTACAGGTGAAGGTCACGTTGG3′ (sgRNA3c), and 5′GTAGCTAAATAAGGGT ATGGAGG3′ (sgRNA4c) with last three nucleotides as PAM sequences. These sgRNAs were fused with tracrRNA sequence, 5′GTTTTAGAGCTAGAAATAGCAAGTTAAA ATAAGGCTAGTCCGTTATCAACTTGAAAAAGTGGCACCGAGTCGGTGCT3′, to form functional guide RNAs.

## Construction of chloroplast genome-editing vectors

For gene expression in chloroplasts, we used the promoters and 5′-untranslated regions of following genes encoded in the *Chlamydomonas* chloroplast genome; *rbcL* (*Franklin et al., 2002*), *psaA* (*Young & Purton, 2014*), and *psbD* (*Kasai et al., 2003*). For terminators of transcription, we used 3′-untranslated regions of the *Chlamydomonas* chloroplast genes, *psbA* (*Noor-Mohammadi, Pourmir & Johannes, 2012*) and *rbcL* (*Young & Purton, 2014*).

For the expression of the selectable marker, *aadA* gene, we used the *rbcL* promoter with its 5′ UTR, and a *psbA* 3′ UTR. The expression cassette was synthesized in three parts by GenScript. Synthesized DNA fragments were assembled using a strategy involving Type IIS restriction enzyme, *Bsa*I (*Bertalan et al., 2015*) and ligated into pBR322 that had been digested with *Hind*III and *Ava*I to create the pBR322-*aadA* intermediate vector. The *aadA* cassette was then transferred to pUC19, resulting in YP4 (Chloroplast Edit Plasmids are listed with the information on their core components in Table 1).

The Cas9 endonuclease gene derived from *Streptococcus pyogenes* (SpCas9) was codon-optimized using the Codon Usage Database (https://www.kazusa.or.jp/codon/; *Nakamura, Gojobori & Ikemura, 2000*) in combination with manual editing to reduce long mononucleotide repeats (see Fig. S1). The Cas9 expression cassette was composed with the *psaA*-exon 1 promoter with its 5′ UTR, the codon-optimized Cas9 (Cas9c) and *rbcL* 3′ UTR. The components were synthesized at GenScript, assembled using *Bsa*I and ligated into YP4. The resulting plasmids, YP5, was used to construct an Edit Plasmid YP11 carrying a cassette for the expression of double sgRNAs.

The sgRNA cassette contained the following elements in 5′–3′ orientation: the 102 bp-long promoter of tRNA^Trp (CCA), tRNA^Trp (CCA), sgRNA (sgRNA1c), tRNA^Lys (UUU), sgRNA (sgRNA2c), tRNA^Leu (UAG) and 101 bp-long terminator of tRNA^Trp (CCA). The second sgRNA cassette with sgRNA (sgRNA3c) and sgRNA (sgRNA4c) was created with the same elements by DNA synthesis followed by *Bsa*I digestion and ligation with YP5 that was digested by *Bgl*II and *Age*I. This resulted in the YP12 construct.

**Table 1  List of Edit Plasmids for organelle transformation.** See 'Materials and Methods' for details.

| Edit Plasmid | Organism | Organelle | Expr cassette 1 promoter for Cas9 | Expr cassette 2 guide RNA | Donor DNA | Vector backbone |
|---|---|---|---|---|---|---|
| YP5 | *Chlamydomonas* | chloroplast | *psaA* | N/A | N/A | pUC19 |
| YP11 | *Chlamydomonas* | chloroplast | *psaA* | sgRNA1c & 2c | N/A | pUC19 |
| YP12 | *Chlamydomonas* | chloroplast | *psaA* | sgRNA3c & 4c | N/A | pUC19 |
| YP13 | *Chlamydomonas* | chloroplast | *psaA* | sgRNA1c & 2c | HR1c:GFPc:HR2c | pUC19 |
| YP14 | *Chlamydomonas* | chloroplast | *psaA* | sgRNA3c & 4c | HR3c:GFPc:HR4c | pUC19 |
| YP6 | *Chlamydomonas* | chloroplast | *psbD* | N/A | N/A | pUC19 |
| YP19 | *Chlamydomonas* | chloroplast | *psbD* | sgRNA1c & 2c | N/A | pUC19 |
| YP20 | *Chlamydomonas* | chloroplast | *psbD* | sgRNA3c & 4c | N/A | pUC19 |
| YP21 | *Chlamydomonas* | chloroplast | *psbD* | sgRNA1c & 2c | HR1c:GFPc:HR2c | pUC19 |
| YP22 | *Chlamydomonas* | chloroplast | *psbD* | sgRNA3c & 4c | HR3c:GFPc:HR4c | pUC19 |
| YP23 | *Chlamydomonas* | chloroplast | N/A | sgRNA1c & 2c | HR1c:GFPc:HR2c | pUC19 |
| YP24 | *Chlamydomonas* | chloroplast | N/A | sgRNA3c & 4c | HR3c:GFPc:HR4c | pUC19 |
| YP29 | *Chlamydomonas* | chloroplast | N/A | sgRNA1c & 2c | HR1c:GFPc:HR2c | pBR322 |
| YP31 | *Chlamydomonas* | chloroplast | *psaA* | sgRNA1c & 2c | N/A | pBR322 |
| YP33 | *Chlamydomonas* | chloroplast | *psbD* | sgRNA1c & 2c | N/A | pBR322 |
| HS6 | *Saccharomyces* | mitochondria | N/A | sgRNA1m & 2m | HR1m:GFPm:HR2m | pBR322 |
| HS8 | *Saccharomyces* | mitochondria | *COX2* | sgRNA1m & 2m | HR1m:GFPm:HR2m | pBR322 |

Next, the donor DNA fragment was synthesized with the GFP gene (GFPc) optimized for the *Chlamydomonas* chloroplast codon usage (Franklin et al., 2002) that was flanked with the sequences homologous to the Cas9 cleavage sites created by the pair of guide RNAs encoded in each construct. Those homologous sequences consisted of 74 bp (HR1) or 76 bp (HR2, HR3, HR4) in length. The synthesized donor fragments were ligated into YP11 and YP12, resulting in YP13 and YP14, respectively.

Another set of Edit Plasmids (YP6, YP19, YP20, YP21, YP22) was constructed similarly with the Cas9 expression cassette that consisted of the *psbD* promoter and its 5′ UTR instead of the *psaA*-exon 1 promoter with its 5′ UTR. Negative control plasmids YP23 and YP24 were constructed from YP13 and YP14, respectively, by removing the Cas9 expression cassette with *Eco*RI and *Hind*III digestion and filling in, followed by ligation. Furthermore, constructs with the pBR322 backbone were made as follows: YP31, by ligating the *Avr*II–*Nhe*I fragment from YP11 and *Nhe*I digested pBR322-*aadA* plasmid; YP33, by ligating the *Avr*II–*Nhe*I fragment from YP19 and *Nhe*I digested pBR322-*aadA* plasmid; YP29, by ligating the *Age*I–*Nhe*I fragment from YP23 and *Age*I/*Nhe*I-digested pBR322-*aadA* plasmid.

## Analysis of donor DNA insertion at Cas9 target sites in *Chlamydomonas* chloroplasts

For PCR analysis of Cas9/gRNA-mediated donor DNA insertions in *Chlamydomonas* chloroplast DNA, cell lysates were prepared from each individual transformant that was grown on TAP/spectinomycin (100 μg/ml) medium, following the protocol described in Barrera, Gimpel & Mayfield (2014). Cell lysates were pooled per construct and subjected to

PCR reactions where each reaction in 25 µl consisted of 2 µl of pooled cell lysates as a template and 2X master mix Q5 high-fidelity polymerase (New England Biolabs, Ipswich, MA, USA). The primers used for PCR amplification were C1 (5′GCTGGTTGGTTCC ACTACCAC3′), C2 (5′CATACGGTGTACAATGTTTCAGTCG3′), C3 (5′CACCTTC AAATTTTACTTCAGCACGTG3′) and C4 (5′GTGAGAAATAATAGCATCACGGT GAC3′). C1 and C4 primers recognized the region of the *Chlamydomonas* chloroplast genome in the vicinity of the designed Cas9/gRNA cleavage sites. C2 and C3 are specific for the donor DNA. The pairs C1–C3 and C2–C4 were used for the amplification of the genomic region replaced with the donor DNA, including the junction regions. The PCR amplification was performed as follows: initial denaturation, 98 °C for 30 s; 35 cycles of 98 °C for 10 s, 56 °C for 20 s and 72 °C for 1 min; final polymerization 72 °C for 2 min. Aliquots of PCR reaction were analyzed on agarose gels. Amplified DNAs were purified using Qiagen PCR Purification kit and subjected to direct sequencing at GENEWIZ, LLC. DNA sequencing analysis was performed by using Vector NTI ContigExpress software (Thermo Fisher Scientific, Waltham, MA, USA).

## Characterization of on-target mutations at Cas9 target site in *Chlamydomonas* chloroplasts

Sequence variations created by Cas9/gRNA activity was analyzed at the *Ava*II recognition site (GGWCC) present within one of the Cas9 target sites, sgRNA2c (5′CTTCACCTG TAAATGGACCA**CGG**3′ PAM sequence in bold).

First, cell lysate was extracted from individual colonies transformed with each construct as described above. The numbers of colonies assayed in this experiment were 10 for YP5 (control), 15 for YP11, 5 for YP29 (control), 10 for YP31 and 7 for YP33 (Step 1). We then made cell lysates from pooled five colonies derived from independent transformation events except for YP33 with seven colonies. Pooled DNA samples were subjected to the amplification of the genomic region containing the target site using the Q5® high-fidelity polymerase-based PCR amplification system (New England Biolabs, Ipswich, MA, USA) and the primer set, C1 and C4 (see above). The PCR reaction was conditioned as follows: initial denaturation, 98 °C for 30 s; 35 cycles of 98 °C for 10 s, 60 °C for 20 s and 72 °C for 30 s; final polymerization 72 °C for 2 min (Step 2). Amplified DNA products were purified using DNA Clean & Concentrator 5 (Zymo Research) and subjected to *Ava*II digestion overnight (Step 3). After gel electrophoresis, the region corresponding to 700–900 bp of each pool, containing undigested DNA of 795 bp, was cut out of agarose gel and the DNA was extracted using QIAquick Gel Extraction Kit (Qiagen, Hilden, Germany) (Step 4). Extracted DNA was then directly cloned into pMiniT2.0 vector according to a manufacturer's protocol (New England Biolabs, Ipswich, MA, USA) (Step 5). We randomly selected 12 *E. coli* colonies from each pool of transformants with positive constructs (YP11, YP31 and YP33) and eight colonies from the pools with control constructs YP5 and YP29, and performed PCR amplification using the primer pair, C1 and C4 (Step 6). Aliquots of PCR reactions were digested with *Ava*II and analyzed by electrophoresis to narrow down the candidates for altered *Ava*II sites through sequencing analysis (Step 7). If a rare DNA editing event occurred, *Ava*II restriction site is

**Table 2 Yeast strains used in this study.**

| Strain | Genotype | Source |
|---|---|---|
| MCC109ρ⁰ | *MATα ade2 ura3 kar1-1* [ρ⁰] | *Costanzo & Fox (1993)* |
| MCC125 | *MATa lys2* [*cox3-10* ρ⁺] | *Costanzo & Fox (1993)* |
| CUY563 | *MATa ade2-101 ade3-24 leu2-3,112 ura3-52* [ρ⁺] | *Roberg et al. (1999)* |
| NB80 | *MATa lys2 leu2-3,112 ura3-52 his3ΔHinDIII arg8::hisG* [ρ⁺] | *Bonnefoy & Fox (2000)* |

likely to be lost, and, thus, *Ava*II digestion would enrich for such events. Selected clones were one from YP5, two from YP11, five from YP31 and three from the YP29 transformants. PCR amplicons were purified and subjected to Sanger-sequencing and analyzed using Vector NTI ContigExpress program (Step 8). In addition, we randomly selected multiple colonies from Step 5, isolated DNA using QIAprep Spin Miniprep kit (Qiagen, Hilden, Germany), sequenced at GENEWIZ, LLC and analyzed for any change in the *Ava*II sequence.

## Strains, media and genetic methods for yeast

The yeast strains used in this study are listed in Table 2. Minimal media and standard genetic methods were described in *Sherman, Fink & Hicks (1986)* unless otherwise specified below.

## Design of sgRNAs for yeast mitochondrial DNA

Guide RNA target sites were selected from the *COX1* gene (Gene ID: 854598) in the mitochondrial genome of *S. cerevisiae S288c* (NCBI Accession: NC_001224). The *COX1* gene encodes cytochrome C oxidase subunit I that is, part of the respiratory electron transport chain in all eukaryotes and is required in yeast for growth on non-fermentable carbon source (https://www.yeastgenome.org/locus/S000007260). To help design and select guide RNA target sites, a web-based Bioinformatics program, CRISPOR (http://crispor.tefor.net/, *Haeussler et al., 2016*) was employed. The unique gRNA targeting sequences chosen for the *COX1* gene were 5′TTCTTTGAAGTATCAGGAGG**TGG**3′ (sgRNA1m) and 5′ATGATTATTGCAATTCCAAC**AGG**3′ (sgRNA2m) with the last 3 nucleotides (bold letters) as PAM sequences. Each of guide RNA target site without PAM sequence was fuzed with tracrRNA to create functional single guide RNAs. The tracrRNA sequence used was:

5′GTTTTAGAGCTAGAAATAGCAAGTTAAAATAAGGCTAGTCCGTTATCAAC TTGAAAAAGTGGCACCGAGTCGGTGCT3′.

## Construction of mitochondrial genome-editing vectors

For the construction of Edit Plasmids to test in yeast mitochondria, pHD6 (*Green-Willms et al., 2001*) was modified as follows. First, pHD6 was digested with *Pst*I and *Hind*III to remove previously cloned fragments except for the following: 0.75 kb *COX3* gene fragment, pBR322-based replication *ori* and *amp*^R. Then, the following elements were inserted: Cas9 expression cassette, sgRNA expression cassette and donor DNA. The sequence encoding Cas9 was codon-optimized for expression in yeast mitochondria

(Fig. S2). In addition, six tryptophan codons (TGG) at positions W18, W464, W659, W883, W1074 and W1126 of the 1366 aa Cas9 protein were changed to TGA, which codes for tryptophan in mitochondria but for a translational stop in the cytoplasm to avoid any Cas9 expression outside of mitochondria. Codon-optimized Cas9 gene (Cas9m) was synthesized along with the 71 bp-long minimal promoter and the 119 bp-long terminator of the *COX2* gene (*Mireau, Arnal & Fox, 2003*), and was flanked with *Pst*I and *Hind*III sites (GenScript). A multicloning locus with *Xba*I, *Not*I and *Nco*I sites was also added 5′ to the *Hind*III site to facilitate subsequent subcloning of other cassettes.

The sgRNA expression cassette contained the following in 5′–3′ orientation: a 75 bp minimal *COX3* promoter consisting of the TATA box, transcription initiation site and 5′UTR; tRNA$^{Phe}$ (GAA); sgRNA2m; a tRNA$^{Trp}$ (UGA); sgRNA1m; tRNA$^{Met}$ (CAU) and a 118 bp minimal *COX3* terminator including the dodecamer (*Turk et al., 2013*). The sgRNA expression cassettes were synthesized with a *Not*I site at the 5′ end and a *Nco*I site at the 3′ end to allow directional cloning in the Edit Plasmid and maintain the direction of transcription of the Cas9 and guide RNA cassettes same on the resulting plasmids.

To construct HS8, the donor DNA carrying the GFP gene was synthesized and cloned into the *Nco*I site of the plasmid that contained Cas9 and guide RNA expression cassettes. The nucleotide sequence encoding GFP was codon-optimized for expression in yeast mitochondria (*Cohen & Fox, 2001*). A tryptophan codon corresponding to W57 was changed to TGA, assuring GFP expression only in mitochondria. The GFP coding region (GFPm) was fused in-frame with the *COX1* ORF at the sgRNA1m cleavage site by adding the following linker sequence between *COX1* ORF (italic lowercase) and GFPm ORF (underlined lowercase): *ttttcggagtttctggtggagg*TGGTGGAatgacacat. Both ends of the GFPm ORF were fused with the *COX1* genomic sequences external to the Cas9 cleavage sites to allow homology-directed recombination: 144 bp adjacent to the sgRNA1m site (HR1m) and 115 bp adjacent to the sgRNA2m site (HR2m). Furthermore, the guide RNA recognition sites within the donor DNA was modified to prevent subsequent cleavages by Cas9: the sgRNA1m sequence was changed at seven out of the 23 nucleotides without altering the encoded amino acid sequence as follows: TTCTTTGAAGTATCAGGAGG TGG (original) to TT**TTT**C**GG**AGT**TT**C**T**GG**T**GG**A**GG (modified nucleotides in donor DNA in bold). The sgRNA2m sequence was modified by deleting 16 nucleotides at the 5′ end of the recognition site as follows: ATGATTATTGCAATTCCAACAGG (original) to CAACAGG.

The control construct HS6 was created by deleting the Cas9 expression cassette from HS8 through the digestion with *Pst*I and *Not*I, followed by electrophoresis and elution of 5.5 kb DNA, and the ligation with the linker (5′GAAGATCTAACCGCGGAA GC3′ and 5′GGCCGCTTCCGCGGTTAGATCTTCTGCA3′).

## Yeast mitochondrial transformation

The constructs were transformed into a yeast line that lacked mitochondrial DNA ($\rho^0$), MCC109$\rho^0$, using the biolistic microprojectile method as described in *Bonnefoy & Fox (2007)*. The transformation was performed together with pYES2 (Invitrogen) as a carrier

plasmid with *URA*3 selectable marker, so that URA$^+$ nuclear transformants could be selected first on minimal medium lacking uracil in supplements. To identify mitochondrial transformants, URA$^+$ colonies were assayed for the ability to rescue the *cox3* deletion mutation of the ρ$^+$ line, MCC125, after the cross. The URA$^+$ transformants that carried Edit Plasmids in mitochondria through co-transformation events were able to provide the wild-type *COX3* fragment to rescue the *cox3* deletion by homologous recombination. The screen was repeated three times to obtain clean lines with Edit Plasmids in mitochondria. To analyze the Cas9 activity of the Edit Plasmids, five isolated lines containing each of Edit Plasmids and the line containing the wild-type mitochondrial genome, CUY563, were activated in YPD medium overnight and crossed together by mixing an aliquot of cells on the SD medium supplemented with adenine and uracil and incubating at 30 °C. Under the condition, transformant and diploid cells could grow but not CUY563, facilitating the cytoplasmic fusion of two strains in the presence of the *kar1-1* mutation, which prevents nuclear fusion and stable diploidization (*Conde & Fink, 1976*). After 48 h of incubation, cells were suspended in 300 μl H$_2$O and concentrated through centrifugation under 3,000×*g* for 5 m and resuspension in 50 μl H$_2$O. One μl of cells was subjected to the analysis of the genome editing effect by Cas9 at the target sites.

## Analysis of donor DNA insertion between two Cas9 target sites and plasmid stability in yeast mitochondria

To detect the presence of the donor DNA inserted between two Cas9 target sites in the mitochondrial genome, two primer sets, C-12 and F-11, were designed to amplify the junction regions: The Primer C (5′CTATTCAGGCACATTCAGGACC3′) recognizing the genomic region 250 bp upstream of the sgRNA1m site and Primer 12 (5′GATCTG CTAATTGTACTGAACCG3′) recognizing the donor DNA 560 bp downstream of the sgRNA1m site, and the Primer 11 (5′CAGGTGAAGGTGAAGGTGATGC3′) recognizing the donor DNA 610 bp upstream of the sgRNA2m site and Primer F (5′AGA GGTATACCAACACAAGATTC3′) recognizing the genomic region 260 bp downstream of the sgRNA2m site. For the PCR amplification, One μl of cell suspension prepared as described above was used for one 25 μl PCR reaction with One-Taq Quick-Load 2X Master Mix (New England Biolabs, Ipswich, MA, USA) and 5 μm of each primer. The PCR amplification was performed as follows: (1) 7 min at 94 °C, (2) 30 s at 94 °C, (3) 30 s at 52 °C, (4) 1 min 30 s at 60 °C, (5) go to step 2 for 39 times, (6) 10 min at 60 °C. The PCR products were separated on 1.2% agarose gel by electrophoresis. The DNA bands that corresponded to the junction fragments were cut out of the gel and subjected to the DNA extraction using the QIAquick Gel Extraction Kit (Qiagen, Hilden, Germany). The eluted DNA samples were sequenced with Primers C, F, 11 and 12 at GENEWIZ, LLC. Vector NTI and SnapGene were used for DNA sequence analyses.

For the analysis of the plasmid stability in yeast mitochondria, the yeast cultures were created as described above with the modification of using the wild-type strain NB80 instead of CUY563 for the cross with the MCC109ρ$^0$ line carrying HS8. The cultures of crossed cells in SD minimal medium were further advanced to the third and fourth overnight culture. Cells were sampled from each culture as described above. The PCR

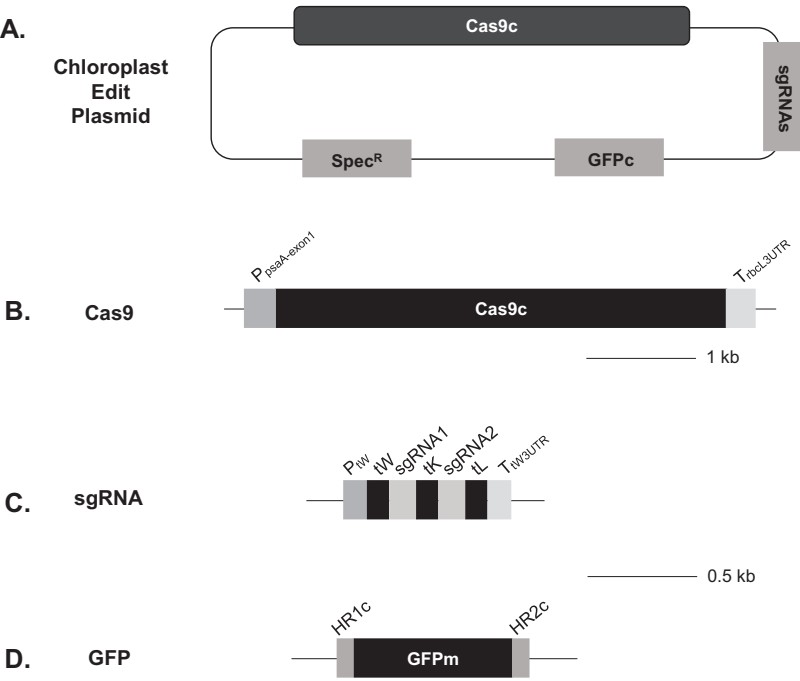

**Figure 1 Schematic representation of the Edit Plasmids for *Chlamydomonas* chloroplasts.** See 'Materials and Methods' for details. (A) Overall structures of Edit Plasmids; (B) Structure of the Cas9 expression cassette; (C) Structure of the gRNA expression cassette; and (D) Structure of the donor DNA. Scales are provided for B–D.

amplification was performed with a primer set, G and 12: Primer G (5′GAGAACAAA TGGTATGACAATGCAT3′) recognizing the donor DNA and Primer 12 (see above). The condition of PCR reaction was same as above.

### Rescue of plasmids from yeast mitochondria transformants

For the rescue of Edit Plasmids from yeast cells, one ml of the second overnight cell culture was sampled and total DNA isolated using the Quick-DNA Miniprep Plus Kit (Zymo Research, Irvine, CA, USA). A total of 200 ng of isolated total DNA was digested with *Apa*I and *Sph*I to cleave pYES2 plasmid DNA in the total DNA fraction, while the HS8 plasmid remained intact as it lacked these restriction sites. After inactivating the restriction enzymes at 65 °C for 20 min, the DNA was transformed into *E. coli* using the Stable competent cells (New England Biolabs, Ipswich, MA, USA). Colonies that grew on LB medium containing carbenicillin were subjected to plasmid DNA isolation. Plasmid DNA was then subjected to digestion with several restriction enzymes, and the digestion products were analyzed by gel electrophoresis.

## RESULTS

### Edit Plasmid design strategy for *Chlamydomonas* chloroplasts

The Edit Plasmids used in the transformation of *Chlamydomonas* chloroplasts were designed and constructed as follows (Fig. 1; Table 1). First, we used either pBR322 or pUC19 for the vector backbone with pMB1 replication origin previously reported to

replicate in chloroplasts (*Boynton et al., 1988*). Second, we optimized the codons of the Cas9 gene derived from *Streptococcus pyogenes* for *Chlamydomonas* chloroplast expression without altering the amino acid sequence and without adding any chloroplast transit peptide sequence or nuclear localization signal sequence (Fig. S1). Third, a strong promoter, *psaA* promoter and its 5′ UTR, and a weaker promoter, *psbD* promoter and its 5′ UTR, from the *Chlamydomonas* chloroplast genome (*Michelet et al., 2011*), were used to express the codon-optimized Cas9 gene. Fourth, four guide RNA sites were selected from the exon 3 of the *psaA* gene to have unique targets in the *Chlamydomonas* chloroplast genome (see Materials and Methods). Fifth, to generate correctly processed gRNAs, tRNA processing scheme was employed as shown by *Xie, Minkenberg & Yang (2015)* with the following modification: Instead of using nuclear RNA polymerase III promoter as described in *Xie, Minkenberg & Yang (2015)* and *Cong et al. (2013)*, we utilized the promoter of tRNA$^{Trp}$ (CCA) gene from the *Chlamydomonas* plastid for the expression of guide RNAs. To have proper processing of sgRNAs following their transcription in chloroplasts, a multiplex sgRNA configuration was designed; tRNA-1, sgRNA-1, tRNA-2, sgRNA-2 and tRNA-3. The tRNA genes of *Chlamydomonas* plastids used for the proper processing were: tRNA$^{Trp}$ (CCA) for tRNA-1, tRNA$^{Lys}$ (UUU) for tRNA-2 and tRNA$^{Leu}$ (UAG) for tRNA-3. Sixth, a donor DNA was designed with the GFPc gene that had been optimized for expression in *Chlamydomonas* chloroplasts (*Franklin et al., 2002*). For homologous recombination of the donor DNA at the sites cleaved by Cas9/double gRNAs, the ends of the donor DNA were fused with the genomic sequence at the cleavage sites in the exon 3 of the *psaA* gene. To preclude additional cleavages by Cas9 and guide RNAs after replacement, HR sequences contained silent mutations at the gRNA target sites such as removing of PAM sequence (see Materials and Methods). In addition, homologous recombination was designed to give an in-frame fusion of GFPc with the *psaA* gene product. Seventh, the *aadA* selectable marker was present on the Edit Plasmid constructs to allow positive selection of the plasmids. The promoter and terminator for the expression of selectable marker were the *rbcL* promoter with its 5′ UTR and the *psbA* 3′ UTR, respectively.

## Cas9/gRNA-driven donor DNA insertion between two Cas9 target sites in *Chlamydomonas* chloroplasts

To assess donor DNA insertion mediated by Cas9/gRNA in *Chlamydomonas* chloroplasts, the wild-type strain of *C. reinhardtii*, CC-125, was transformed with the following Edit Plasmids, YP13, YP14, YP21, or YP22 (see Materials and Methods, Table 1). These plasmids were designed to evaluate an optimal promoter strength for Cas9 expression and to compare the efficacy of gRNAs (sgRNA1c–sgRNA2c vs. sgRNA3c–sgRNA4c). Promoters used were a strong promoter (*psaA* promoter with its 5′ UTR) and a weaker promoter (*psbD* promoter with its 5′ UTR). In addition, two negative control plasmids, YP23 and YP24, carrying two gRNAs and donor DNA but no Cas9, were included to assess the background frequency of spontaneous homologous recombination.

For the screening of Cas9-mediated donor DNA insertion events among transformants, we designed the PCR primer pairs which consisted of a primer specific to the chloroplast

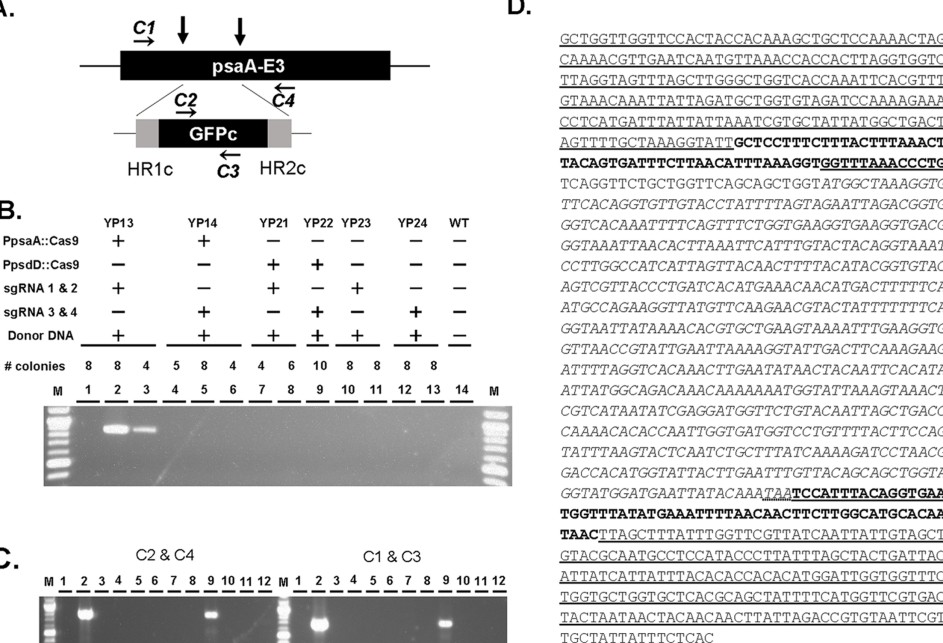

**Figure 2 Replacement of the chloroplast genome with donor DNA by Cas9/gRNA.** (A) Schematic view of the *psaA-E3* genomic region targeted by two gRNAs (vertical arrows) and the donor DNA composed of codon-optimized GFPc gene that is provided by the Edit Plasmid. The recognition sites of primers used for the amplification of the junction regions are indicated (C1–C4). (B) PCR amplification of the junction region of the replaced DNA. Pooled DNAs extracted from independent colonies was used as templates with primers C2 and C4. Total number of colonies for each Edit Plasmids was 20 for YP13, 17 for YP14, 10 for YP21, 10 for YP22, 16 for YP23 and 24. Template for Lane 14 was untransformed wild-type cells. C2/C4 amplicon was 852 bp long. M: 1 kb plus molecular weight marker (New England Biolabs, Ipswich, MA, USA). (C) De-convolution of junction-PCR positive pools of YP13 transformants. A total of 12 events of the positive pools from Lanes 2 and 3 of (B) were analyzed by two primer sets C2/C4 and C1/C3 to amplify the left and right junction regions, respectively. Events #2 and #9 carried replaced DNA. C1/C3 amplicon was 712 bp long. (D) The sequence obtained from PCR amplification of the replacement DNA locus in *Chlamydomonas* plastid DNA modified by the Edit Plasmid approach. Underlined sequences: wild-type chloroplast genomic sequence that are not present on the Edit Plasmid. Sequences in bold: homologous regions (HR1c and HR2c) present in the donor DNA on the Edit Plasmid. Sequences in bold underlined: modified gRNA target sites present in the donor DNA. Sequences with double underlines: Silent mutations at the 3′ side of guide RNA sites to preclude re-cleavage by Cas9/sgRNA. (D) The sequence obtained from PCR amplification of the replacement DNA locus in *Chlamydomonas* plastid DNA modified by the Edit Plasmid approach. Underlined sequences: wild-type chloroplast genomic sequence that are not present on the Edit Plasmid. Sequences in bold: homologous regions (HR1c and HR2c) present in the donor DNA on the Edit Plasmid. Sequences in bold underlined: modified gRNA target sites present in the donor DNA. Sequences with dotted underlines: silent mutations at the 3′ side of guide RNA sites to preclude re-cleavage by Cas9/sgRNA.

genome at the integration sites and a primer specific to GFPc (Fig. 2A). The primer sets did not produce any DNA product after PCR amplification using the DNA from the wild-type strain as template (Fig. 2B, lane 14).

A total of 28 days post transformation, cell lysates from 20 independent colonies of all constructs except for YP23 and YP24 with 16 colonies were prepared in pools. The lysates were used for analyzing the replacement events with donor DNA through PCR amplification using a primer pair C2 and C4 (see Materials and Methods). PCR screening

**Table 3 Summary of Cas9/gRNA induced integration of donor DNA into organelle genomes.** Numbers in parentheses: Independent transgenic events tested in *Chlamydomonas* chloroplasts and biological replications tested in yeast mitochondria.

| Construct | Organism | Organelle | Cas9 | Cas9 promoter | sgRNA | Donor DNA | Vector | Donor DNA integration |
|-----------|----------|-----------|------|---------------|-------|-----------|--------|----------------------|
| YP13 | *Chlamydomonas* | chloroplast | + | *psaA* | 1c/2c | + | pUC19 | + (2/20) |
| YP14 | *Chlamydomonas* | chloroplast | + | *psaA* | 3c/4c | + | pUC19 | − (0/17) |
| YP21 | *Chlamydomonas* | chloroplast | + | *psbD* | 1c/2c | + | pUC19 | − (0/10) |
| YP22 | *Chlamydomonas* | chloroplast | + | *psbD* | 3c/4c | + | pUC19 | − (0/10) |
| YP23 | *Chlamydomonas* | chloroplast | − | − | 1c/2c | + | pUC19 | − (0/16) |
| YP24 | *Chlamydomonas* | chloroplast | − | − | 3c/4c | + | pUC19 | − (0/16) |
| HS6 | *Saccharomyces* | mitochondria | − | − | 1m/2m | + | pBR322 | − (0/15) |
| HS8 | *Saccharomyces* | mitochondria | + | COX2 | 1m/2m | + | pBR322 | + (15/15) |

showed that the expected sizes of the amplicon was only detected from the pools derived from YP13 transformants (Fig. 2B, lanes 2 and 3), but not from other constructs including negative control constructs, YP23 and YP24, lacking the Cas9 expression cassette but having donor DNA as well as gRNAs (Fig. 2B, lanes 10, 11, 12 and 13; Table 3). This suggested that there was no detectable homologous recombination using Edit Plasmids containing donor DNA without Cas9 activity. This observation is consistent with the result observed by *Dauvillee et al. (2004)*, showing that the lengths of the left and right homologous regions used in this experiment (74 bp and 76 bp, respectively) were insufficient to induce homologous recombination in *Chlamydomonas* chloroplasts effectively. Based on these control results, we conclude that the positive detection of donor DNA insertion events in the YP13 construct was induced by Cas9/guide RNA activity.

YP13 contained two guide RNAs, gRNA1c and gRNA2c, and the strong *psaA* promoter with its 5′UTR for the Cas9 expression. YP14 with different gRNAs, gRNA3c and gRNA4c, with other components same as YP13 failed to detect replacement events (Fig. 2B, lanes 4, 5 and 6; Table 3). This contrasting result showed the varying efficacy of designed guide RNAs. The other contrasting result was observed in the strength of promoters driving Cas9. Edit Plasmid YP21, in which Cas9 was expressed under the weaker promoter, *psbD* promoter in combination with the efficacious gRNAs (gRNA1c and gRNA2c), did not yield any expected amplicon (Fig. 2B, lanes 7 and 8; Table 3).

Further PCR with individual cell lysates of positive pools of YP13 transformants was performed for de-convolution (Fig. 2C). Two out of 12 transformants taken from the two pools showed amplicons of the expected size using primer pair C1/C3 as well as primer pair C2/C4, representing the left and right junction fragments. Overall, the frequency of positive events was two out of 20 independent transformants (10%) with the YP13 construct that contained the two guide RNAs, sgRNA1c and sgRNA2c, and Cas9 expressed by the strong *psaA* promoter. In experiments involving the other constructs that did not contain these elements, for example, constructs with Cas9 expressed by the weak promoter, it was zero out of 69 independent transformants (0%) in total (see Table 3). The *t*-test of the data set gave *p*-value of 0.022 and *z*-score of 2.8, confirming the significance of the positive events.

Amplified DNA fragments were sequenced to further confirm successful donor DNA insertion at Cas9 target sites. Sequence analysis revealed that the entire length of donor DNA was integrated in the *Chlamydomonas* chloroplast DNA (Fig. 2D). The genomic sequence corresponded to the expected sequence from insertion of the donor DNA at the two Cas9 cleavage sites. Precise donor DNA insertion was observed in *Chlamydomonas* chloroplasts by the use of an Edit Plasmid that encoded a Cas9 expression cassette, a multiple guide RNA expression cassette and a donor DNA template.

## Analysis for on-target mutations at one Cas9/gRNA target site in *Chlamydomonas* chloroplasts

A number of reports showed that Cas9/gRNA induced cleavages in the nucleus are often repaired through (NHEJ) pathways, resulting in nucleotide substitutions (SNPs), small insertions and small deletions (INDELs) (*Li et al., 2015*; *Shin et al., 2016*). In order to survey whether such mutations occur through Cas9/gRNA in chloroplasts, we constructed several Edit Plasmids that carried sgRNA2c in the guide RNA expression cassette besides Cas9 (YP11, YP31 and YP33; see Table 1). We utilized sgRNA2c as its target site (5′CTTCACCTGTAAATGGACCACGG3′, Cas9 cleavage site in bold with PAM sequence in the last three nucleotides) harbored the *Ava*II restriction sequence (*Ava*II restriction site is GGWCC where W is either A or T) (Fig. 3). Thus, any change such as base substitution and INDELs at the Cas9 cleavage site would result in the loss of *Ava*II restriction. We used the loss of *Ava*II restriction to enrich for rare gene edited sequences. This approach of enriching for mutations to a restriction site was used successfully to recover rare Cas9-mediated NHEJ events in nuclear DNA in several previous reports (*Jiang et al., 2014*; *Xie & Yang, 2013*; *Xie, Minkenberg & Yang, 2015*). The control constructs were YP5 and YP29 without the guideRNA or Cas9, respectively. The cells transformed with these constructs were subjected to DNA extraction and subsequent PCR amplification of the genomic region of the target site using primers C1 and C4, and the high fidelity Q5 DNA polymerase (see Fig. 2A and Materials and Methods). The amplified DNA samples were digested with *Ava*II and separated through gel electrophoresis. There was no clearly visible DNA band on the agarose gel that corresponded to the undigested DNA specific to positive samples, indicating that mutations at the cleavage site occurred infrequently (see Supplemental Information). To detect rare events of such mutations, we attempted to isolate DNA from the region (700–900 bp) including undigested DNA (795 bp), while wild-type DNA fragments produced by *Ava*II digestion were 329 bp and 469 bp long. Extracted DNA fractions were subjected to cloning into an *E. coli* vector. Transformed *E. coli* colonies were sampled randomly and their DNA samples were further analyzed by *Ava*II digestion and/or direct sequencing (see Fig. 3; Table 4). The analysis of 66 clones in total from all Edit Plasmids carrying active Cas9 under strong *psaA* promoter (YP11 and YP31) showed that four independent clones (6.06%) had SNPs resulting in the loss of the *Ava*II site. The analysis of 100 clones lacking Cas9 (YP5) or guide RNAs (YP29), representing negative controls, showed that seven independent clones (7%) had SNPs resulting in the loss of the *Ava*II site. No INDEL mutations were detected in this experiment.

```
                                        PAM     AvaII
        parent       TTA GAA GCT CAC CGT GGT CCA TTT ACA GGT GAA G
                      L   E   A   H   R   G   P   F   T   G   E
        YP11a        TTA GAA GCT CAC CGT GGC CCA TTT ACA GGT GAA G
                      L   E   A   H   R   G   P   F   T   G   E
        YP11b        TTA GAA GCT CAC CGT GGT TCA TTT ACA GGT GAA G
                      L   E   A   H   R   G   S   F   T   G   E
        YP31a        TTA GAA GCT CAC CGT GGT TCA TTT ACA GGT GAA G
                      L   E   A   H   R   G   S   F   T   G   E
        YP31b        TTA GAA GCT CAC CGT GGC CCA TTT ACA GGT GAA G
                      L   E   A   H   R   G   P   F   T   G   E
        YP5a, control TTA GAA GCT CAC CGT GGC CCA TTT ACA GGT GAA G
                      L   E   A   H   R   G   P   F   T   G   E
        YP5b, control TTA GAA GCT CAC CGT GAT CCA TTT ACA GGT GAA G
                      L   E   A   H   R   D   P   F   T   G   E
        YP5c, control TTA GAA GCT CAC CGT GAT CCA TTT ACA GGT GAA G
                      L   E   A   H   R   D   P   F   T   G   E
        YP5d, control TTA GAA GCT CAC CGT GGT CAA TTT ACA GGT GAA G
                      L   E   A   H   R   G   Q   F   T   G   E
        YP29a, control TTA GAA GCT CAC CGT GGT TCA TTT ACA GGT GAA G
                      L   E   A   H   R   G   S   F   T   G   E
        YP29b, control TTA GAA GCT CAC CGT GGT TCA TTT ACA GGT GAA G
                      L   E   A   H   R   G   S   F   T   G   E
```

**Figure 3 Detection of SNPs at Cas9/sgRNA2c cleavage site in *Chlamydomonas* chloroplasts.** DNA Sequences (5′–3′) near the cleavage site in cloned amplicons lacking the *Ava*II site are aligned with the wild-type parent DNA shown on the top line. Deduced amino acid sequence is shown under each DNA sequence. The 20-nucleotide target sequence for the Cas9/sgRNA complex is indicated in blue. The PAM site (in green), the *Ava*II recognition site (in bold blue), and SNPs and the resulting amino acid changes (in red) are also labeled.

**Table 4 Summary of DNA alterations at sgRNA2c site in *Chlamydomonas* chloroplasts.** SNP frequency was measured at the *Ava*II site in chloroplast transformants with Edit Plasmids carrying different Cas9 promoters (*psaA*, *psbD* or none), vector backbone, and with and without the guide RNA (sgRNA2c) or donor DNA. SNP frequency was deduced by the number of SNP detected per number of amplicon clones analyzed (see 'Materials and Methods').

| Experiment type | Construct | Cas9 | Cas9 promoter | sgRNA 2c | Donor | Vector | SNP freq. | % |
|---|---|---|---|---|---|---|---|---|
| Test | YP11 | + | *psaA* | + | − | pUC19 | 2/48 | |
| | YP31 | + | *psaA* | + | − | pBR322 | 2/18 | |
| | Total: YP11 + LP31 | + | *psaA* | + | − | | 4/66 | 6.06 |
| | YP33 | + | *psbD* | + | − | pBR322 | 0/24 | |
| | Total: YP11 + LP3 + YP33 | + | | + | − | | 4/90 | 4.44 |
| Negative control | YP5 | + | *psaA* | − | − | pUC19 | 5/49 | |
| | YP29 | + | − | + | + | pBR322 | 2/51 | |
| | Total: YP5 + YP29 | + | | | | | 7/100 | 7.00 |

## Construction of Edit Plasmids for yeast mitochondria

It has previously been shown that plasmids derived from pBR322 were capable of replicating in yeast mitochondria (*Fox, Sanford & McMullin, 1988*). To create Edit Plasmid constructs for yeast mitochondria, we used pBR322 as the backbone for the study of Cas9/gRNA-driven genome editing in mitochondria. As shown in Fig. 4, the Edit Plasmid constructs contained the following elements similar to the Edit Plasmids for *Chlamydomonas* chloroplasts: Cas9 expression cassette, guide RNA expression cassette

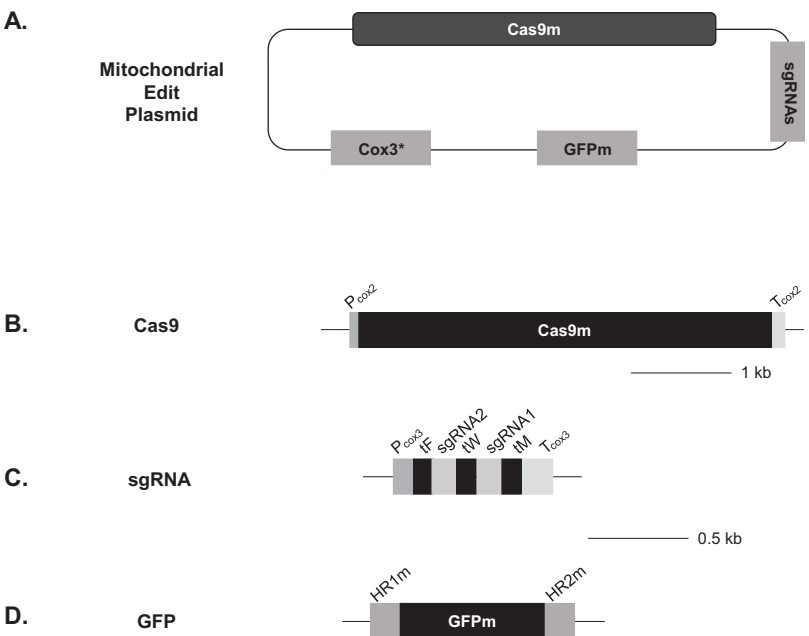

**Figure 4 Schematic representation of the Edit Plasmids for yeast mitochondria.** See "Materials and Methods" for details. (A) Overall structures of Edit Plasmids; (B) Structure of the Cas9 expression cassette; (C) Structure of the gRNA expression cassette; and (D) Structure of the donor DNA. Scales are provided for B–D.

and donor DNA for DNA insertion at Cas9 target sites. In addition, the *COX3* fragment (0.75 kb *Pac*I-*Mbo*I). was used as a screenable marker for mitochondrial transformation with its capability to rescue the *COX3* deletion mutant *cox3-10* as described in *Fox, Sanford & McMullin (1988)*. The Cas9 expression cassette had the Cas9 coding sequence that was optimized for expression in yeast mitochondria (Fig. S2). As part of codon optimization, the codons that were not used at all or were rarely used in yeast mitochondria were replaced with codons that were used frequently. Also, a number of TGG were replaced with UGA, which is a stop codon in the universal codon system but is translated into tryptophan in yeast mitochondria (*Fox, 1979*). This was designed to prevent expression of Cas9 in the cytoplasm after microprojectile DNA transformation. The minimal promoter and terminator of *COX2* gene were used to drive expression of the Cas9 gene. The minimal lengths of 75 bp and 119 bp of these elements were chosen with the purpose of suppressing spontaneous homologous recombination at the sites and avoiding integration into the mitochondrial genome as shown in *Mireau, Arnal & Fox (2003)*.

For the guide RNA target sites in yeast mitochondria, we chose the *COX1* gene, encoding subunit I of cytochrome C oxidase required for oxidative phosphorylation, that is, respiration. Two sites unique to the mitochondrial genome were selected: 5′TTCTTT GAAGTATCAGGAGGTGG3′ (sgRNA1m) and 5′ATGATTATTGCAATTCCAACA GG3′ (sgRNA2m) with the last 3 nucleotides as PAM sequences. The sgRNA1m site was localized in exon 4 of the *COX1* gene, and the sgRNA2m in exon 5. The two sites are separated by 1,258 bp encompassing an intron (Fig. 5). Each of these guide

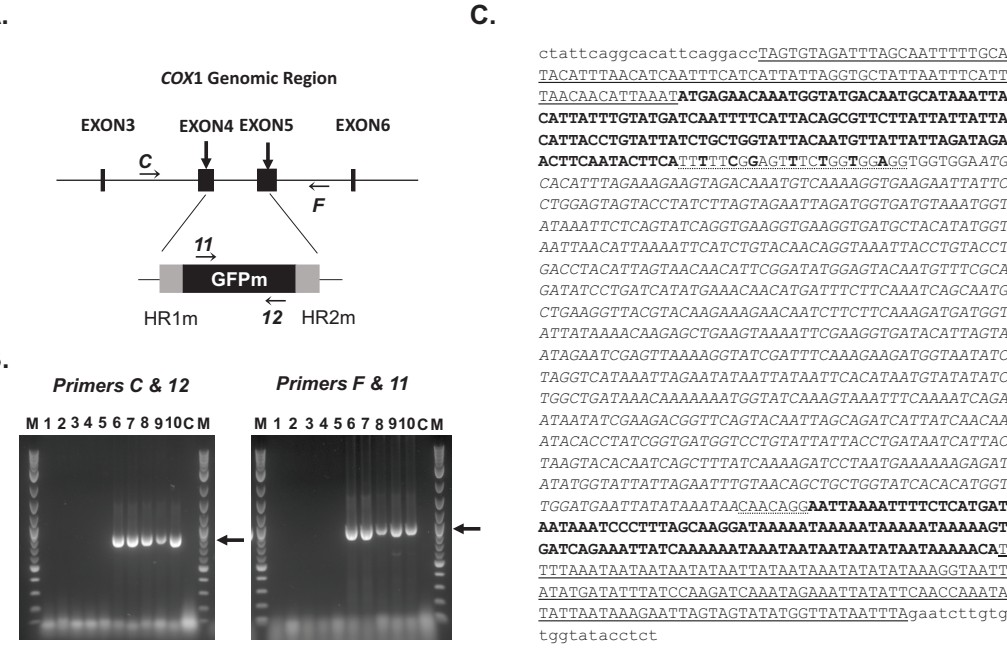

**Figure 5 Replacement of the mitochondrial genome with donor DNA by Cas9/gRNA.** (A) Schematic view of the *COX1* genomic region targeted by two gRNAs (arrows) and the donor DNA with GFP gene that is provided by the Edit Plasmid. The recognition sites of primers used for the amplification of the junction regions are indicated (C, F, 11 and 12). (B) PCR analysis of the junction regions of the integrated donor DNA. Left: 5′ region amplified with C/12 primer set; right: 3′ region amplified with F/11 primer set. Lanes 1–5: Five control lines with HS6 Edit Plasmid without Cas9 activity; lanes 6–10; Five lines with HS8 Edit Plasmid with Cas9 activity. (C) Wild-type CUY563 strain; (M) 1 kb plus molecular weight marker (New England Biolabs, Ipswich, MA, USA). Arrows: the size expected from the deduced sequence with integrated donor DNA (C/12 amplicon: 870 bp; F/11 amplicon: 907 bp). DNA fragments separated in lanes 6 and 10 for the both amplicons were isolated for sequence confirmation (see text). (C) The sequence obtained from PCR amplification of the replacement DNA locus in transformed yeast mitochondrial DNA modified by the Edit Plasmid approach. Underlined sequences: wild-type mitochondrial genomic sequences that are not present on the Edit Plasmid. Sequences in bold: short homologous regions present in the donor DNA (HR1 and HR2) adjacent to gRNA target sites. Sequences with dotted underlining: modified gRNA target sites present in the donor DNA (altered nucleotides are shown in bold). The codon-optimized GFP coding region is presented in italics. Sequences presented in lower case correspond to primers C and F that were used for amplification of the replacement DNA locus. Homologous recombination leading to DNA replacement occurred without causing any sequence changes either in the replacement DNA nor in the surrounding wild-type mitochondrial DNA.

RNA sequence without PAM sequence was fuzed with the tracrRNA sequence (see Materials and Methods). For the guide RNA expression, the gRNA cassette was constructed in 5′–3′ orientation as follows: a minimal *COX3* promoter; tRNA$^{Phe}$ *(GAA)*; sgRNA2m; tRNA$^{Trp}$ *(UCA)*; sgRNA1m; tRNA$^{Met}$ *(CAU)* and a minimal *COX3* terminator element (Fig. 4). For the donor DNA insertion experiment, the donor DNA contained the GFPm gene that was codon-optimized for expression in yeast mitochondria (*Cohen & Fox, 2001*). In addition, one tryptophan codon was changed to UGA, further assuring GFPm expression only in mitochondria. The GFPm coding region in the donor DNA was designed to be in-frame with the *COX1* ORF after the donor DNA insertion. Further,

the both ends of the donor DNA were fused with the *COX1* genomic sequences at the external junction of the Cas9 cleavage sites. HR1 and HR2 corresponded to two short genomic regions which were each immediately adjacent to the corresponding guide RNA target site (see Fig. 4). The length of the homologous region at each end was chosen to be relatively short to minimize spontaneous homologous recombination without Cas9 activity, that is, 144 bp adjacent to the sgRNA1m site (HR1m), 115 bp adjacent to the sgRNA2m site (HR2m), similar to those shown in *Mireau, Arnal & Fox (2003)*. Furthermore, sequence variations were included at the guide RNA recognition sites within the donor DNA, so that the replaced mitochondrial DNA would no longer be recognized by the guide RNA/Cas9 system. Seven of the 20 nucleotides in the sgRNA1m recognition site have been changed, and the sgRNA2m site has been changed by deleting 16 nucleotides at the 5′ end of the recognition site (see Materials and Methods). This design was to prevent the deletion of replaced DNA from further genome editing activity and increase the stability of replaced DNA in the presence of the Edit Plasmids.

## Cas9/gRNA-driven donor DNA insertions at Cas9 target sites in yeast mitochondria

To assay for donor DNA insertion events through Cas9-induced cleavages, the construct HS8 and its control construct HS6 lacking Cas9 were each transformed into a strain lacking mitochondrial DNA ($\rho^0$) as described in "Materials and Methods." HS8 and HS6, both carried the guide RNA expression cassette and donor DNA with GFPm. The only difference was the presence and absence of the Cas9 expression cassette, respectively. Transformants with each of mitochondrial constructs were identified by subsequent screening for their capability to rescue the *cox3* deletion mutant (see Materials and Methods). The isolated mitochondrial transformants were then crossed with the strain CUY563, carrying the wild-type mitochondrial genome, to observe the effect of Edit Plasmids on the mitochondrial genome. Five transformant lines of each construct were subjected to the cross with the wild-type strain on a glucose medium. After 48 h of incubation, cells were collected and assayed for the donor DNA insertion events by PCR amplification (see Materials and Methods). Primer sets were used wherein one primer was from the mitochondrial *COX1* genomic region in the vicinity of the cleavage sites and the other primer was from the donor DNA region (see Fig. 5A). This design of primers was to ensure that the desired PCR product could only be amplified from a correctly replaced DNA in the mitochondrial genome but not from the wild-type mitochondrial DNA nor from the Edit Plasmid itself. As shown in Fig. 5B, the expected size of the DNA amplicons from each end of the replaced DNA was observed in the HS8 derived cell culture. No DNA fragments were amplified from the culture transformed with the control construct HS6. To further assess the reproducibility of the data, we repeated the same experiment 15 times starting from the cross between Edit Plasmid lines and wild-type yeast strain, CUY563. The donor DNA insertion was detected from all 15 HS8 samples containing Cas9 but none from 15 HS6 samples lacking Cas9. Then, the amplified DNA fragments from three independent crosses between HS8 and CUY563

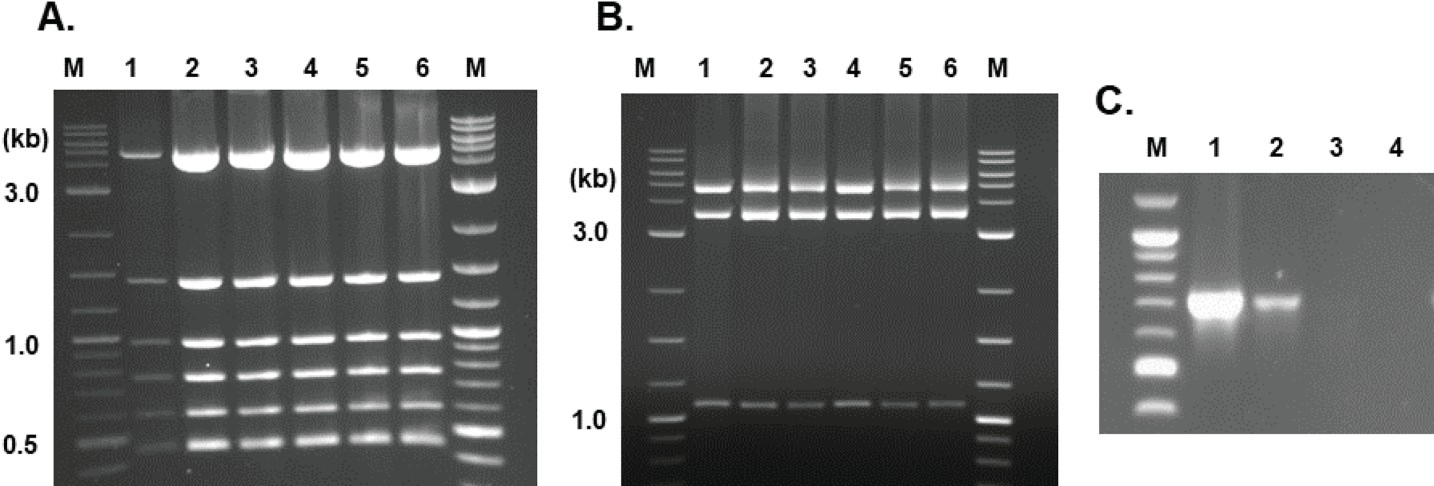

**Figure 6 Characterization of Edit Plasmids rescued from yeast mitochondrial transgenic cells.** (A and B) Restriction analysis of rescued Edit Plasmids. Lane 1, the original Edit Plasmid DNA, Lane 2–6, rescued Edit Plasmids from yeast cells. Plasmids were digested with *Nco*I and *Cla*I (A) or *Nco*I and *Pst*I (B), and separated by 1.0% agarose gel electrophoresis. (C) Semi-quantitative PCR assay of the Edit Plasmid DNA (HS8) after the cross with the wild-type ρ⁺ strain. Lane 1, first overnight culture; lane 2, second; lane 3, third; and lane 4, fourth overnight culture. (M) 1 kb plus molecular weight marker (New England Biolabs, Ipswich, MA, USA).                               

strains were sequenced directly. Sequence analysis revealed that all amplicon sequences covering the replaced region completely matched that of the donor DNA present in HS8 construct (Fig. 5C; Supplemental Information). These results showed that complete donor DNA insertion was successfully induced in yeast mitochondria by the use of an Edit Plasmid that encodes Cas9 and guide RNA expression cassettes and a donor DNA template.

## Rescue of the Edit Plasmid out of yeast mitochondria and its stability

To confirm the presence of HS8 plasmid after the cross with the wild-type mitochondria, we isolated the intact plasmid out of crossed yeast cells. For the plasmid rescue experiment, we sampled cells from the second overnight culture where cells have gone through at least 20 generations after the cross. Since the Edit Plasmid line was co-transformed with the nuclear plasmid, pYES2, we digested the total DNA isolated from the cultured cells with *Apa*I and *Sph*I to eliminate pYES2 from the rescue. Transforming 200 ng of total DNA isolated from the yeast cells into *E. coli*, we obtained 12 colonies. All of them showed restriction patterns indistinguishable from that of the transforming HS8 construct by two sets of digestions (Figs. 6A and 6B). This showed that the Edit Plasmid was able to persist even in the presence of the wild-type mitochondrial DNA.

To address the question of Edit Plasmid stability, we advanced the second culture of crossed cells to two additional overnight cultures. The presence of the HS8 Edit Plasmid in the series of cultures was analyzed by sampling cells with primer set G-12 to amplify part of the Edit Plasmid, that is, the donor DNA. As shown in Fig. 6C, the donor DNA was detected in the first two cultures in a diminishing manner and it was not detectable in the last two cultures. This demonstrates that the Edit Plasmid tends to cease its presence in mitochondria in the presence of wild-type mitochondrial DNA within about 30–40 generations.

## DISCUSSION

### A plasmid approach enabling organelle genome editing

The CRISPR/Cas9 system is a powerful tool to precisely edit genomes of a wide range of prokaryotic and eukaryotic species. However, gene editing in eukaryotes, including yeast *S. cerevisiae* and algal *C. reinhardtii*, has been restricted to nuclear genomes. There is no published report except one that describes the use of the system on organelle genomes. The sole publication reports that the Cas9 protein expressed in cytoplasm is also localized in human mitochondria, and the simultaneous expression of Cas9 and guide RNA in the cytoplasm resulted in the reduction of the target gene in mitochondria (*Jo et al., 2015*). While this is an encouraging report, there is no analysis on the change on DNA caused by Cas9 at the nucleotide level. Thus, it is difficult to conclude if the reported change in the overall gene dosage was directly caused by Cas9 induced cleavages or not. Further, if it were direct, the approach would be still limited to the creation of gene knockouts but not to the replacement of DNA with donor DNA because DNA could not be freely imported into mitochondria.

Here we show an original approach that is, capable of editing and replacing mitochondrial as well as chloroplast DNA in yeast and alga, respectively. Our approach utilizes plasmids that persist in organelles at least long enough to edit organelle genomes. pBR322 plasmid was reported to replicate in yeast mitochondria (*Fox, Sanford & McMullin, 1988*) and chloroplasts of *Chlamydomonas* (*Boynton et al., 1988*). The plasmids that were equipped with genome editing elements, which we called Edit Plasmids, were transformed into mitochondria of a yeast, *S. cerevisiae*, and chloroplasts of an alga, *C. reinhardtii*, to test the universal functionality of our approach.

### Cas9 induced homology-directed repair through donor DNA insertion in organelles

Organelles contains their genomes in multiple copies. There are about 50 copies of mitochondrial DNA in yeast (review: *Williamson, 2002*) and about 83 copies in *Chlamydomonas* chloroplasts (*Gallaher et al., 2018*). The genomes in high copy numbers are postulated to be required for keeping the genetic content uniform through active homologous recombination among them. Indeed, active homologous recombination was utilized for transgenic approaches to integrate transgenic DNA into the organelle genomes. For our approach, Edit Plasmids were designed not to integrate into genomes. This autonomous feature is highly desirable for producing the edited cells and organisms without carrying any transgenic elements left in their genomes after the Edit Plasmids are segregated away. For this purpose, we designed expression elements within Edit Plasmids as short as possible to reduce the chance of integration. For instance, promoter and terminator elements to drive Cas9 expression in yeast mitochondria had lengths of 74 bp and 119 bp, respectively. The same elements for guide RNA expression in yeast mitochondria had 75 bp and 118 bp, respectively. For reference, conventional integration experiments in organelles were usually induced by longer DNA homology to the target site such as fragments with lengths of 1.6 kb and 2.3 kb (*Cohen & Fox, 2001*).

The other precautious measure that we included in the Edit Plasmids in this study is to differentiate the homologous recombination facilitated by Cas9-induced cleavage from spontaneous recombination without Cas9. For this purpose, we also reduced the length of the homologous regions flanking the donor DNA to about 100 bp or less. As noted above, donor DNA insertion was detected in yeast mitochondria from all 15 HS8 samples containing Cas9 but none from 15 HS6 samples lacking Cas9. Since the control constructs without Cas9 produced no detectable replacement of donor DNA in the organelle genomes, our designs were shown sufficient to accomplish the task. If homologous recombination would have occurred by a single crossover event in the absence of DNA cleavage, the entire Edit Plasmid would have integrated at one of the two homologous regions. Such a product would have been detectable in one of the PCR amplicons of our experiments. Since no such products were detected, we conclude that our experimental setting was successful in preventing spontaneous homologous recombination by either single or double crossover events.

As for the DNA repair system of double-strand breaks in organelles, a number of studies showed that homology-directed repairs (HDR) occur often to preserve genetic information without causing any mutation (review: *Stein & Sia, 2017*). Therefore, we presumed that DNA double strand breaks induced by Cas9 will be predominantly repaired by homologous recombination with uncut wild-type DNA. Since the repaired DNA should be identical to the wild-type DNA, the analysis of such events would not be easily measurable. To make the Cas9 activity measurable, we focused on experiments to detect donor DNA insertion events at the region that were deleted by two cleavage sites induced by Cas9. The donor DNA fragments were so designed that the guide RNA target sites were eliminated after the homologous recombination to stabilize the replacement events.

As summarized in Table 3, the results in *Chlamydomonas* chloroplasts revealed that only the construct with Cas9 under the strong promoter *psaA* and sgRNA1c/2c (YP13) showed Cas9/sgRNA activity, that is, donor DNA integration by HDR-mediated gene replacement, while the rest, including two controls (YP23 and YP24), did not. Both the strength of promoters driving Cas9 and the efficacy of the sgRNAs influence Cas9/sgRNA activity. In chloroplasts, Cas9 under a strong promoter (*psaA*) but not under a weak promoter (*psbD*) and sgRNA pair 1c/2c but not sgRNA3c/4c showed Cas9/sgRNA activity. If Cas9-independent homologous recombination would have occurred spontaneously in the two observed events with the strong *psaA* promoter, we would have expected to see single-crossover events at one of the homologous regions as the predominant products. This would have resulted in the integration of the entire Edit Plasmid at the recombination site and the PCR amplification of only one of the two ends of the donor DNA integration. Since both ends of the donor DNA were confirmed in the two events by PCR analysis, we conclude that they were the products of double-crossover events at two cleavage sites produced by Cas9. The absence of Cas9-independent recombination events in our experiment is also supported by the fact that we did not observe any integration of either end of the donor DNA among 69 transformants of other constructs, for example, constructs carrying Cas9 driven by a weak promoter. The sequencing of

the integration events further confirmed that Cas9-dependent homologous recombination is the primary mode of the precise donor DNA integration.

With regard to the length of homologous DNA to induce recombination, *Dauvillee et al. (2004)* reported no DNA insertion in *Chlamydomonas* chloroplast above background with the homology shorter than 100 bp, irrespective of the length of homology at the other end. Examining $2 \times 10^7$ cells, they reported that the shortest homology regions capable of recombination were 75 bp at one end and 121 bp at the other end. Although we did not calculate number of cells in our experiments, we estimate it to be significantly less than $10^5$. Thus, we detected targeted donor DNA insertions at a much higher frequency and with much shorter length of homology at the both ends (74 bp and 75 bp) than the experiments previously reported to be insufficient for recombination insertion in *Chlamydomonas* chloroplasts (*Dauvillee et al., 2004*). In comparison, we detected no donor DNA insertion in 69 transplastomic control lines lacking functional Cas9 or sgRNA. Similarly, donor DNA insertion in yeast mitochondria was confirmed by PCR analysis and DNA sequencing. In this case, the donor DNA was flanked by the short length of homologous regions at the both ends (115 bp and 144 bp) to suppress spontaneous recombination. Consequently, no insertion of the donor DNA was detected in the absence of Cas9 activity while the insertion was detected in the presence of Cas9 activity in mitochondria.

Although we confirmed the presence of intact GFP gene in both mitochondria and chloroplasts, we were unable to detect GFP fluorescence in either. In yeast, the failure of the detection could be due to the fusion of GFP with the 224 amino acid long, amino-terminal fragment of COX1 protein that is, known to contain five transmembrane domains (https://www.uniprot.org/uniprot/A0A0H3WI17). The presumed integration into the inner membrane of mitochondria might have prevented GFP protein from folding properly and being active. In the case of *Chlamydomonas*, the direct detection of GFP signals in cells was known to be challenging (*Franklin et al., 2002*). Besides, the target site we selected was in the *psaA* gene which was known to be trans-spliced with remaining parts of mRNA to produce the full-length mRNA for protein synthesis in chloroplasts (*Choquet et al., 1988*). It remains to be answered whether or not the insertion of GFP donor DNA affected the proper trans-splicing of the RNA transcribed from the recombined region.

Despite the lack of GFP signals, these experimental designs resulted in the demonstration of functional Cas9 activity introduced by the Edit Plasmids in two distinct organelles in two distinct organisms. Integration of donor DNA at the Cas9 target sites was complete without leaving any nucleotide alteration. To our knowledge, this is the first demonstration of Cas9 induced donor DNA insertions mediated by Cas9/guideRNAs in mitochondria and chloroplasts. Organelle genomes have constitutions and environments distinct from the nucleus. For example, some mitochondrial genomes are low in GC content (17% in yeast mitochondria; *Foury et al., 1998*) and pH milieu is different from the other part of cells (pH 7.5 in yeast mitochondria whereas pH 7.2 in the cytoplasm; *Orij et al., 2009*). Our results further demonstrate that the CRISPR/Cas9 system is functional in both nuclear and organelle genomes. We found no evidence of Cas9 toxicity in stably

transformed yeast and algal organelles unlike that previously reported for nuclear expression of Cas9 expression in *Chlamydomonas* (*Jiang et al., 2014*).

## No evidence of Cas9 induced INDEL mutations at a Cas9 target site in *Chlamydomonas* chloroplasts

Double-stranded DNA breaks induced by Cas9/sgRNA can be repaired either by HDR or by non-homologous end joining (NHEJ) pathways in the nucleus. The latter often results in mutations, including small insertions, deletions, or substitutions at or close to cleavage sites (*Brinkman et al., 2018*). We showed the former repair system is active in organelles as discussed above. We further explored possible repairs of dsDNA breaks by NHEJ by analyzing one of the Cas9 target sites that harbored a site recognized by a DNA restriction enzyme. We made use of the site to screen any change created by NHEJ repairs. Enriching such changes by *Ava*II enzyme digestion, gel separation, cloning the *Ava* II-resistant amplicon, and sequencing the cloned amplicon, we detected four events of nucleotide substitutions out of 90 (4.44%) clones with enriched amplicons from the cells transformed with Cas9/gRNA active constructs (YP11, YP31 and YP33) (Table 4). Considering the *psbD* promoter is insufficient for Cas9 activity as we showed above, the frequency of substitutions in relevant constructs (YP11 and YP31) is 4/66 = 6.06% whereas the control constructs (YP5 and YP29) had seven nucleotide substitutions out of 100 enriched clones. The results indicate that the SNPs detected at the *Ava*II site in experimental lines and control lines are statistically similar and not a consequence of Cas9-induced SNPs. Enrichment of mutations at the *Ava*II recognition site is not due to Cas9 activity since they are found in similar frequency in the control lines without Cas9 or sgRNA. They likely represent enrichment of preexisting native polymorphisms or PCR errors at the *Ava*II site by our strategy for enriching the *Ava*II resistant clones. Based on our data, we did not find evidence for NHEJ DNA repair in chloroplasts. This is consistent with prior reports in *Chlamydomonas* chloroplasts and in mitochondria of higher plants, showing the lack of NHEJ repair events (*Odom et al., 2008*; *Kazama et al., 2019*). In the latter, the engineered TALEN targeted to mitochondria was used to create a specific DNA cleavage. Through thorough molecular analyses, they demonstrated that the cleavage was not repaired by NHEJ mechanisms but by homology-dependent recombination. Similar findings were reported by others, discussing the absence of the conserved KU proteins involved in NHEJ repair in *Chlamydomonas* chloroplasts (*Kwon, Huq & Herrin, 2010*). As most bacteria have low or no NHEJ repair systems (*Gomaa et al., 2014*; *Citorik, Mimee & Lu, 2014*; *Su et al., 2016*), the lack of NHEJ in organelles may reflect their prokaryotic ancestry in evolution. In Cas9 based genome editing in nuclear genomes, INDELs are the most common mutations observed at cleavage sites and are advantageous for knocking out genes encoded in the nucleus (*Li et al., 2015*; *Shin et al., 2016*). Our results reveal that such knockout mutations are difficult to obtain in organelles. When such mutagenesis is required in organelles, an optimal approach is to introduce specific substitutions through donor DNA insertion at the site induced by two cleavages as we showed in this report.

### Edit Plasmids maintain their autonomy in organelles

The autonomous nature of the Edit Plasmids is beneficial for their application in organelle genome editing. It ensures an easy removal of transgenic components of Cas9 and guide RNAs after editing is completed. It also allows the complete removal of Cas9 activity by a simple segregation. We provide evidence that the Edit Plasmids persist long enough to induce gene editing. To demonstrate the persistence of the Edit Plasmid, we performed the isolation of the Edit Plasmid out of yeast cells. The cells were grown over 20 generations after crossing the Edit Plasmid and the wild-type mitochondrial genome together. We could successfully retrieve intact Edit Plasmid. Without any positive selection, Edit Plasmids tend to segregate away after many generations as shown for yeast mitochondria in this study. To prolong the activity of Edit Plasmids and to obtain the homoplasmic state of engineered DNA, we plan to test mitochondrial *rep/ori* elements that are more efficient than the one carried on pBR322 or have a selectable marker on the Edit Plasmid, such as *ARG8m* for yeast (*Bonnefoy & Fox, 2000*), to enhance their stability. Our goal is to extend this technology to crop plants without the use of a selectable marker for regulatory considerations and public acceptance.

## CONCLUSIONS

Organelle genome editing has been underexplored in comparison with that of nuclear genes. We report the first demonstration of Cas9-based genome editing in organelles. Our novel approach utilizing Edit Plasmids was successful in inducing donor DNA insertion at the target sites cleaved by Cas9/gRNAs in yeast mitochondria as well as in algal chloroplasts. The frequency of substitutions and INDEL mutations at a cleavage site was below detection in our study, indicating that homology-directed DNA repair & replacement are the main outcomes of Cas9 induced cleavages in organelles. The approach with Edit Plasmids is expected to open the door to edit organelle DNA precisely as well as to introduce new alleles and genes without leaving any trace of transgenes. Future experiments will address the following: increasing the efficiency of organelle genome editing via CRISPR/Cas9; and the use of CRISPR/Cas9 to digest unmodified organelle DNA and promote homoplasmy. Possible application areas of the organelle genome editing are wide, impacting agriculture, industrial biotechnology and human healthcare.

## ACKNOWLEDGEMENTS

We thank Dr. Thomas Fox for providing yeast strains and other materials, and Victoria Kuhnel for her assistance in laboratory work.

### Funding

This work was supported by the internal funds available at Napigen, Inc. The funders had no role in study design, data collection and analysis, decision to publish, or preparation of the manuscript.

## Grant Disclosures

The following grant information was disclosed by the authors:
Napigen, Inc.

## Competing Interests

Hajime Sakai is CEO at Napigen. Byung-Chun Yoo, Narendra Yadav and Emil Orozco, Jr are employees at Napigen. All authors are co-inventors on patent application US2019/0136249 A1 "organelle genome modification using polynucleotide guided endonuclease," submitted by Napigen. The authors declare that they have no other competing interests.

## Author Contributions

- Byung-Chun Yoo analyzed the data, conceived and designed the experiments, performed the experiments, prepared figures and/or tables, authored or reviewed drafts of the paper, and approved the final draft.
- Narendra S. Yadav analyzed the data, conceived and designed the experiments, performed the experiments, prepared figures and/or tables, authored or reviewed drafts of the paper, and approved the final draft.
- Emil M. Orozco Jr analyzed the data, conceived and designed the experiments, prepared figures and/or tables, and approved the final draft.
- Hajime Sakai analyzed the data, conceived and designed the experiments, performed the experiments, prepared figures and/or tables, authored or reviewed drafts of the paper, and approved the final draft.

## DNA Deposition

The following information was supplied regarding the deposition of DNA sequences:

Cas9c (codon-optimized for *Chlamydomonas* chloroplasts) and Cas9m (codon-optimized for yeast mitochondria) are available as Figs. S1 and S2. The sequences are available at GenBank: MK048157 and MK048158.

## Data Availability

The raw data are available as Supplemental Files.

## Supplemental Information

Supplemental information for this article can be found online at http://dx.doi.org/10.7717/peerj.8362#supplemental-information.

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

# PeerJ

**Barrera D, Gimpel J, Mayfield S. 2014.** Rapid screening for the robust expression of recombinant proteins in algal plastids. *Methods in Molecular Biology* **1132**:391–399 DOI 10.1007/978-1-62703-995-6_26.

**Bertalan I, Munder MC, Weiß C, Kopf J, Fischer D, Johanningmeier U. 2015.** A rapid, modular and marker-free chloroplast expression system for the green alga *Chlamydomonas reinhardtii*. *Journal of Biotechnology* **195**:60–66 DOI 10.1016/j.jbiotec.2014.12.017.

**Bonnefoy N, Fox TD. 2000.** In vivo analysis of mutated initiation codons in the mitochondrial *COX2* gene of *Saccharomyces cerevisiae* fused to the reporter gene *ARG8(m)* reveals lack of downstream reinitiation. *Molecular and General Genetics* **262(6)**:1036–1046 DOI 10.1007/PL00008646.

**Bonnefoy N, Fox TD. 2007.** Directed alteration of *Saccharomyces cerevisiae* mitochondrial DNA by biolistic transformation and homologous recombination. *Methods in Molecular Biology* **372**:153–166 DOI 10.1007/978-1-59745-365-3_11.

**Boynton JE, Gillham NW, Harris EH, Hosler JP, Johnson AM, Jones AR, Randolph-Anderson BL, Robertson D, Klein TM, Shark KB, Sanford JC. 1988.** Chloroplast transformation in *Chlamydomonas* with high velocity microprojectiles. *Science* **240(4858)**:1534–1538 DOI 10.1126/science.2897716.

**Brinkman EK, Chen T, De Haas M, Holland HA, Akhtar W, Van Steensel B. 2018.** Kinetics and fidelity of the repair of Cas-9-induced double-strand DNA breaks. *Molecular Cell* **70(5)**:801–813 DOI 10.1016/j.molcel.2018.04.016.

**Choquet Y, Goldschmidt-Clermont M, Girard-Bascou J, Kück U, Bennoun P, Rochaix J-D. 1988.** Mutant phenotypes support a *trans*-splicing mechanism for the expression of the tripartite *psaA* gene in the *C. reinhardtii* chloroplast. *Cell* **52(6)**:903–913 DOI 10.1016/0092-8674(88)90432-1.

**Citorik RJ, Mimee M, Lu TK. 2014.** Sequence-specific antimicrobials using efficiently delivered RNA-guided nucleases. *Nature Biotechnology* **32(11)**:1141–1145 DOI 10.1038/nbt.3011.

**Cohen JS, Fox TD. 2001.** Expression of green fluorescent protein from a recoded gene inserted into *Saccharomyces cerevisiae* mitochondrial DNA. *Mitochondrion* **1(2)**:181–189 DOI 10.1016/S1567-7249(01)00012-5.

**Conde J, Fink GR. 1976.** A mutant of *Saccharomyces cerevisiae* defective for nuclear fusion. *Proceedings of the National Academy of Sciences of the United States of America* **73(10)**:3651–3655 DOI 10.1073/pnas.73.10.3651.

**Cong L, Ran FA, Cox D, Lin S, Barretto R, Habib N, Hsu PD, Wu X, Jiang W, Marraffini LA, Zhang F. 2013.** Multiplex genome engineering using CRISPR/Cas systems. *Science* **339(6121)**:819–823 DOI 10.1126/science.1231143.

**Costanzo MC, Fox TD. 1993.** Suppression of a defect in the 5′ untranslated leader of mitochondrial *COX3* mRNA by a mutation affecting an mRNA-specific translational activator protein. *Molecular and Cellular Biology* **13(8)**:4806–4813 DOI 10.1128/MCB.13.8.4806.

**Cummins PL, Kannappan B, Gready JE. 2018.** Directions for optimization of photosynthetic carbon fixation: ruBisCO's efficiency may not be so constrained after all. *Frontier in Plant Science* **9**:183 DOI 10.3389/fpls.2018.00183.

**Dauvillee D, Hilbig L, Preiss S, Johanningmeier U. 2004.** Minimal extent of sequence homology required for homologous recombination at the *psb*A locus in *Chlamydomonas reinhardtii* chloroplasts using PCR-generated DNA fragments. *Photosynthesis Research* **79(2)**:219–224 DOI 10.1023/B:PRES.0000015384.24958.a9.

**Estavillo GM, Crisp PA, Pornsiriwong W, Wirtz M, Collinge D, Carrie C, Giraud E, Whelan J, David P, Javot H, Brearley C, Hell R, Marin E, Pogson BJ. 2011.** Evidence for a SAL1-PAP

chloroplast retrograde pathway that functions in drought and high light signaling in *Arabidopsis*. *Plant Cell* **23(11)**:3992–4012 DOI 10.1105/tpc.111.091033.

**Foury F, Roganti T, Lecrenier N, Purnelle B. 1998.** The complete sequence of the mitochondrial genome of *Saccharomyces cerevisiae*. *FEBS Letters* **440(3)**:325–331 DOI 10.1016/S0014-5793(98)01467-7.

**Fox TD. 1979.** Five TGA "stop" codons occur within the translated sequence of the yeast mitochondrial gene for cytochrome c oxidase subunit II. *Proceedings of the National Academy of Sciences of the United States of America* **76(12)**:6534–6538 DOI 10.1073/pnas.76.12.6534.

**Fox TD, Sanford JC, McMullin TW. 1988.** Plasmids can stably transform yeast mitochondria lacking endogenous mtDNA. *Proceedings of the National Academy of Sciences of the United States of America* **85(19)**:7288–7292 DOI 10.1073/pnas.85.19.7288.

**Franklin S, Ngo B, Efuet E, Mayfield SP. 2002.** Development of a GFP reporter gene for *Chlamydomonas reinhardtii* chloroplast. *Plant Journal* **30(6)**:733–744 DOI 10.1046/j.1365-313X.2002.01319.x.

**Gallaher SD, Fitz-Gibbon ST, Strenkert D, Purvine SO, Pellegrini M, Merchant SS. 2018.** High-throughput sequencing of the chloroplast and mitochondrion of *Chlamydomonas reinhardtii* to generate improved de novo assemblies, analyze expression patterns and transcript speciation, and evaluate diversity among laboratory strains and wild isolates. *Plant Journal* **93(3)**:545–565 DOI 10.1111/tpj.13788.

**Gomaa AA, Klumpe HE, Luo ML, Selle K, Barrangou R, Beisel CL. 2014.** Programmable removal of bacterial strains by use of genome-targeting CRISPR-Cas systems. *mBio* **5(1)**:e00928-13 DOI 10.1128/mBio.00928-13.

**Green-Willms NS, Butler CA, Dunstan HM, Fox TD. 2001.** Pet111p, an inner membrane-bound translational activator that limits expression of the *Saccharomyces cerevisiae* mitochondrial gene *COX2*. *Journal of Biological Chemistry* **276(9)**:6392–6397 DOI 10.1074/jbc.M009856200.

**Haeussler M, Schönig K, Eckert H, Eschstruth A, Mianné J, Renaud J-B, Schneider-Maunoury S, Shkumatava A, Teboul L, Kent J, Joly J-S, Concordet J-P. 2016.** Evaluation of off-target and on-target scoring algorithms and integration into the guide RNA selection tool CRISPOR. *Genome Biology* **17(1)**:148–159 DOI 10.1186/s13059-016-1012-2.

**Hanson MR, Bentolila S. 2004.** Interactions of mitochondrial and nuclear genes that affect male gametophyte development. *Plant Cell* **16(Suppl. 1)**:S154–S169 DOI 10.1105/tpc.015966.

**Ishiga Y, Watanabe M, Ishiga T, Tohge T, Matsuura T, Ikeda Y, Hoefgen R, Fernie AR, Mysore KS. 2017.** The SAL-PAP chloroplast retrograde pathway contributes to plant immunity by regulating glucosinolate pathway and phytohormone signaling. *Molecular Plant-Microbe Interactions* **30(10)**:829–841 DOI 10.1094/MPMI-03-17-0055-R.

**Jaganathan D, Ramasamy K, Sellamuthu G, Jayabalan S, Venkataraman G. 2018.** CRISPR for crop improvement: an update review. *Frontier in Plant Science* **9**:985 DOI 10.3389/fpls.2018.00985.

**Jiang W, Brueggeman AJ, Horken KM, Plucinak TM, Weeks DP. 2014.** Successful transient expression of Cas9 and single guide RNA genes in *Chlamydomonas reinhardtii*. *Eukaryotic Cell* **13(11)**:1465–1469 DOI 10.1128/EC.00213-14.

**Jin S, Daniell H. 2015.** The engineered chloroplast genome just got smarter. *Trends in Plant Science* **20(10)**:622–640 DOI 10.1016/j.tplants.2015.07.004.

**Jo A, Ham S, Lee GH, Lee Y-I, Kim SS, Lee Y-S, Shin J-H, Lee Y. 2015.** Efficient mitochondrial genome editing by CRISPR/Cas9. *BioMed Research International* **2015**:1–10 DOI 10.1155/2015/305716.

**Kasai S, Yoshimura S, Ishikura K, Takaoka Y, Kobayashi K, Kato K, Shinmyo A. 2003.** Effect of coding regions on chloroplast gene expression in *Chlamydomonas reinhardtii*. *Journal of Bioscience and Bioengineering* **95(3)**:276–282 DOI 10.1016/S1389-1723(03)80029-4.

**Kazama T, Okuno M, Watari Y, Yanase S, Koizuka C, Tsuruta Y, Sugaya H, Toyoda A, Itoh T, Tsutsumi N, Toriyama K, Koizuka N, Arimura S-I. 2019.** Curing cytoplasmic male sterility via TALEN-mediated mitochondrial genome editing. *Nature Plants* **5(7)**:722–730 DOI 10.1038/s41477-019-0459-z.

**Kwon T, Huq E, Herrin DL. 2010.** Microhomology-mediated and nonhomologous repair of a double-strand break in the chloroplast genome of Arabidopsis. *Proceedings of the National Academy of Sciences of the United States of America* **107**:3954–13959 DOI 10.1073/pnas.1004326107.

**Leslie M. 2018.** 'Old' genome editors might treat mitochondrial diseases. *Science* **361(6409)**:1302 DOI 10.1126/science.361.6409.1302.

**Li Z, Liu Z-B, Xing A, Moon BP, Koellhoffer JP, Huang L, Ward RT, Clifton E, Falco SC, Cigan AM. 2015.** Cas9-guide RNA directed genome editing in soybean. *Plant Physiology* **169(2)**:960–970 DOI 10.1104/pp.15.00783.

**McBride KE, Svab Z, Schaaf DJ, Hogan PS, Stalker DM, Maliga P. 1995.** Amplification of a chimeric *Bacillus* gene in chloroplasts leads to an extraordinary level of an insecticidal protein in tobacco. *Biotechnology* **13**:362–365 DOI 10.1038/nbt0495-362.

**Michelet L, Lefebvre-Legendre L, Burr SE, Rochaix J-D, Goldschmidt-Clermont M. 2011.** Enhanced chloroplast transgene expression in a nuclear mutant of *Chlamydomonas*. *Plant Biotechnology Journal* **9(5)**:565–574 DOI 10.1111/j.1467-7652.2010.00564.x.

**Mireau H, Arnal N, Fox TD. 2003.** Expression of Barstar as a selectable marker in yeast mitochondria. *Molecular Genetics and Genomics* **270(1)**:1–8 DOI 10.1007/s00438-003-0879-2.

**Morley SA, Nielsen BL. 2016.** Chloroplast DNA copy number changes during plant development in organelle DNA polymerase mutants. *Frontiers in Plant Science* **7(480)**:57 DOI 10.3389/fpls.2016.00057.

**Nakamura Y, Gojobori T, Ikemura T. 2000.** Codon usage tabulated from international DNA sequence databases: status for the year 2000. *Nucleic Acids Research* **28(1)**:292 DOI 10.1093/nar/28.1.292.

**Nielsen AZ, Mellor SB, Vavitsas K, Wlodarczyk AJ, Gnanasekaran T, De Jesus MPRH, King BC, Bakowski K, Jensen PE. 2016.** Extending the biosynthetic repertoires of cyanobacteria and chloroplasts. *Plant Journal* **87(1)**:87–102 DOI 10.1111/tpj.13173.

**Noor-Mohammadi S, Pourmir A, Johannes TW. 2012.** Method to assemble and integrate biochemical pathways into the chloroplast genome of *Chlamydomonas reinhardtii*. *Biotechnology and Bioengineering* **109(11)**:2896–2903 DOI 10.1002/bit.24569.

**Odom OW, Baek K-H, Dani RN, Herrin DL. 2008.** *Chlamydomonas* chloroplasts can use short dispersed repeats and multiple pathways to repair a double-strand break in the genome. *Plant Journal* **53(5)**:842–853 DOI 10.1111/j.1365-313X.2007.03376.x.

**Oey M, Lohse M, Kreikemeyer B, Bock R. 2009.** Exhaustion of the chloroplast protein synthesis capacity by massive expression of a highly stable protein antibiotic. *Plant Journal* **57(3)**:436–445 DOI 10.1111/j.1365-313X.2008.03702.x.

**Orij R, Postmus J, Beek AT, Brul S, Smits GJ. 2009.** In vivo measurement of cytosolic and mitochondrial pH using a pH-sensitive GFP derivative in *Saccharomyces cerevisiae* reveals a relation between intracellular pH and growth. *Microbiology* **155(1)**:268–278 DOI 10.1099/mic.0.022038-0.

**Ramesh VM, Bingham SE, Webber AN. 2011.** A simple method for chloroplast transformation in *Chlamydomonas reinhardtii*. *Methods in Molecular Biology* **684**:313–320 DOI 10.1007/978-1-60761-925-3_23.

**Roberg KJ, Crotwell M, Espenshade P, Gimeno R, Kaiser CA. 1999.** *LST1* is a *SEC24* homologue used for selective export of the plasma membrane ATPase from the endoplasmic reticulum. *Journal of Cell Biology* **145(4)**:659–672 DOI 10.1083/jcb.145.4.659.

**Sherman F, Fink GR, Hicks JB. 1986.** *Methods in yeast genetics*. Cold Spring Harbor: Cold Spring Harbor Laboratory.

**Shin S-E, Lim J-M, Koh HG, Kim EK, Kang NK, Jeon S, Kwon S, Shin W-S, Lee B, Hwangbo K, Kim J, Ye SH, Yun J-Y, Seo H, Oh H-M, Kim K-J, Kim J-S, Jeong W-J, Chang YK, Jeong B-R. 2016.** CRISPR/Cas9-induced knockout and knock-in mutations in *Chlamydomonas reinhardtii*. *Scientific Reports* **6(1)**:27810 DOI 10.1038/srep27810.

**Stein A, Sia EA. 2017.** Mitochondrial DNA repair and damage tolerance. *Frontiers in Bioscience* **22**:920–943 DOI 10.2741/4525.

**Su T, Liu F, Gu P, Jin H, Chang Y, Wang Q, Liang Q, Qi Q. 2016.** A CRISPR-Cas9 assisted non-homologous end-joining strategy for one-step engineering of bacterial genome. *Scientific Reports* **6(1)**:37895 DOI 10.1038/srep37895.

**Svab Z, Hajdukiewicz P, Maliga P. 1990.** Stable transformation of plastids in higher plants. *Proceedings of the National Academy of Sciences of the United States of America* **87(21)**:8526–8530 DOI 10.1073/pnas.87.21.8526.

**Turk EM, Das V, Seibert RD, Andrulis ED. 2013.** The mitochondrial RNA landscape of *Saccharomyces cerevisiae*. *PLOS ONE* **8(10)**:e78105 DOI 10.1371/journal.pone.0078105.

**Viitanen PV, Devine AL, Khan MS, Deuel DL, Van Dyk DE, Daniell H. 2004.** Metabolic engineering of the chloroplast genome using the *Echerichia coli ubiC* gene reveals that chorismite is a readily abundant plant precursor for p-hydroxybezoic acid biosynthesis. *Plant Physiology* **136(4)**:4048–4060 DOI 10.1104/pp.104.050054.

**Williamson D. 2002.** The curious history of yeast mitochondrial DNA. *Nature Review Genetics* **3**:475–481 DOI 10.1038/nrg814.

**Xie K, Minkenberg B, Yang Y. 2015.** Boosting CRISPR/Cas9 multiplex editing capability with the endogenous tRNA-processing system. *Proceedings of the National Academy of Sciences of the United States of America* **112(11)**:3570–3575 DOI 10.1073/pnas.1420294112.

**Xie K, Yang Y. 2013.** RNA-guided genome editing in plants using a CRISPR-cas system. *Molecular Plant* **6(6)**:1975–1983 DOI 10.1093/mp/sst119.

**Young RE, Purton S. 2014.** Cytosine deaminase as a negative selectable marker for the microalgal chloroplast: a strategy for the isolation of nuclear mutations that affect chloroplast gene expression. *Plant Journal* **80(5)**:915–925 DOI 10.1111/tpj.12675.