# Peer review of "Cas9/gRNA-mediated genome editing of yeast mitochondria and Chlamydomonas chloroplasts"

_PeerJ, doi:10.7717/peerj.8362_

## Round 0.1 · original submission · Major Revisions

Your manuscript "Cas9/gRNA-mediated chloroplast and mitochondrial genome editing" has been assessed by our reviewers. Although it is of interest, we are unable to consider it for publication in its current form. The reviewers have raised a number of points which we believe would improve the manuscript and may allow a revised version to be published in PeerJ. If you are able to fully address these points, we would encourage you to submit a revised manuscript to PeerJ. I look forward to receiving your revised manuscript soon.

Reviewer 1 ·

Basic reporting

Manuscript reports on DNA engineering in Chlamydomonas chloroplasts and yeast mitochondria using an independently replicating episomal element as an expression platform. Authors selected replicons that have been described in the literature. The so called Edit Plasmids carry the expected genes: Cas9, sgRNAs processed by tRNAs, the selectable plastid or mitochondrial marker, and a GFP flanked by short target sequences for insertion at the target site. The chloroplast data are more comprehensive, and with the addition of relatively small amount of data can make a convincing case for CRISPR/Cas action. The mitochondrial data are sketchy.

Experimental design

Problems with the experimental design and suggestion to improve the text are included in general comments.

Validity of the findings

See in general comments.

Additional comments

1. Introduction: Lines 69-71: “Examples include DNA transformation in yeast mitochondria (Fox, Sanford & McMullin, 1988) and Chlamydomonas chloroplasts (Boynton et al., 1988), and high protein expression in tobacco chloroplasts (McBride et al., 1995).” To be equitable, first transformation should be cited for higher plant chloroplasts also.

2. What authors refer to as chloroplast DNA replacement is GFP insertion at the target site facilitated by homologous recombination. DNA replacement would be replacing one gene coding region (or sequence) with an altered coding sequence, such as one psaA coding sequence with a different psaA coding sequence. Make changes throughout.

3. Data presentation should be improved in the text. On Lines 399-400 authors state: “Twenty eight days post transformation, cell lysates from 20 independent colonies of all constructs except for YP23 and YP24 with 16 colonies were prepared in pools.” In reality, the pool sizes were different. When accounting for lanes in Figure 2B, the caption states “Numbers of transgenic colonies were 8 for Lane 1 & 2, 4 for Lane 3, 5”. Thus, for YP13 three pools were analyzed; two pools of 8 and one pool of 4. The pool size could be included in Figure 2B, and the lengthy explanation deleted from the caption.

4. Ultimately, two GFP carrying colonies were found in 20 transplastomic events, and none in 69 “controls” where CRISPR/Cas9 was ineffective or absent, suggesting that the two homologous recombination events were triggered by cleavage of the target site by CRISPR/Cas9. Apparently, when the target site is cleaved, shorter homology is sufficient to facilitate homologous recombination. These are the only data supporting the claim that insertion is due to CRSPR/Cas action, and are not the products of homologous recombination mediated via the short targeting sequences. Authors argue that the sequences are too short to yield transplastomic events. However, Dauville et al. (2004) report on average 2 and 3 events per 2x107 cells using 75 bp/52 bp and 51 bp/52 bp homologies, respectively (Table 2 in reference). The homologous regions in Dauville et al. (2004) are shorter than the 74 bp and 76 bp used in present study. One would like to see shorter homologies and larger sample sizes tested to be comfortable with the conclusion that insertion is due to CRISPR/Cas action.

5. Insertion was into the psaA gene coding region is generating a fusion protein. Are the Chlamydomonas cells expressing the fusion protein photoautotrophic? Was the medium set up to accommodate photoheterotrophic cells? Why were the GFP insertion colonies not screened by fluorescence under UV light?

6. Line 433 Section title: “Cas9/gRNA-induced NHEJ DNA repair in Chlamydomonas chloroplasts”. The title should be changed, since there is no evidence for DNA repair by NHEJ. The section title should be “On target mutations at the Cas9 target site”. It appears that there was positive selection against recovering products of NHEJ. The target sites are within two photosynthetic genes, psaA and psbD. CRISPR/Cas9 most often causes single nucleotide insertions/deletions generating an early stop codon. If the Chlamydomonas cells were grown on photoautotrophic medium, only nucleotide substitutions would be compatible with photoautotrophic growth because during photoautotrophic grown there is strong selection to maintain functional psaA and psbD copies. Indeed, only nucleotide conversions were obtained, that would suggest functional constrains. Discuss the impact of photoautotrophic / photoheterotrophic growth on the outcome of the DNA repair experiment.

6. There are extensive data on chloroplast DNA repair after cleavage by the I-CreII homing endonuclease (Kwon T et al. PNAS 2010;107:13954-13959). Should be cited and discussed. In Arabidopsis, psbA defective sectors survived because were supported by wild-type sectors enabling detection of NHEJ.

7. The statistics of CRISPR/Cas9 mutations is debatable: 4 events in 100 clones with CRISPR/Cas9 and 2 events (based on 1 event in 50 clones) in the absence of CRISPR/Cas9.

8. Mitochondrial replicons (HS8, HS6, etc) should be listed in a Table, as the chloroplast vectors in Table 1.

9. Data for mitochondrial replicons should be included in a Table, similar to Table 3A for chloroplasts.

10. Line 512: DNA replacement is a misnomer – should be DNA insertion. See Point 2 for chloroplasts above.

11. Since GFP was flanked by homologous targeting sequences, the role of CRISPR/Cas in the insertion is questionable. See point 4 for chloroplasts above.

12. Supplemental Figure S1 is apparently missing. To avoid the problem, all Supplemental Figures and Tables should be included as a single pdf file. Raw data should be in separate folder with a list of all documents systematically labelled.

Reviewer 2 ·

Basic reporting

This manuscript demonstrates chloroplast genome editing in Chlamydomonas reinhardtii and mitochondrial genome editing in Saccharomyces cerevisiae, mediated by CRISPR-Cas9 system with the Edit plasmids. This is the first report of CRISPR-based organelle genome editing; therefore, this study harbors potentially sufficient impact and importance for publication in PeerJ. However, there are a lot of points that need to be clarified, improved, and discussed before publication.

Experimental design

See general comments.

Validity of the findings

See general comments.

Additional comments

1) Two key experiments are missing in this study. The first point that needs to be addressed is the percentages of edited organelles in the selected clones. If there was only a small portion of edited organelles in the whole population of clonal cells, the authors' approach is not practically useful. This point should be examined by out-out PCR (i.e., primers C1 and C4 in psaA targeting, and primers C and F in COX1 targeting). The second critical point is stability of the edited organelles. The authors have to conduct long-term monitoring of the edited clones, and the proportion of edited organelles should be assessed over time.

2) Related to the above point, the clonal analysis of yeast mitochondrial genome editing is definitely needed.

3) The manuscript title should include organism names for clarification. The following title should be more appropriate: "Cas9/gRNA-mediated Genome Editing of Yeast Mitochondria and Chlamydomonas Chloroplasts."

4) The authors should explain why they selected the exon 3 of psaA gene as a target site for chloroplast genome editing and what the function of psaA gene is.

5) Scale bars should be added to Figures 2A and 5A.

6) Fluorescence images should be shown, because GFP genes were knocked-in in chloroplast and mitochondrial genomes.

7) Full sequences of the Edit plasmids with the annotations should be provided to secure the reproducibility.

8) Ideally, various lengths of homology arms should be tested and their comparative analyses should be performed. This point should at least be discussed as a future study.

9) Appropriate reference for the previous studies reporting mitochondrial genome editing in humans using ZFNs and TALENs should be added and discussed (Bacman et al., Nature Medicine, 2013; Bacman et al., Nature Medicine, 2018; Gammage et al., Nature Medicine, 2018).

Reviewer 3 ·

Basic reporting

This is a strong and convincing report on successful editing of the organellar (both chloroplast and mitochondrial) DNAs. The research was completed using plasmids introduced into the organelles using biolistics, and the plasmids carry the coding regions for the guide RNAs and cas9 protein. The experiments were carried out in yeast for the mitochondrion, which makes sense as amodel system because of the existence of mitochondrial mutants in this fungus, and the algae Chamdymonas, which has one big chloroplast and is therefore a good model system.

The manuscript is extremely well written and clear.

One potentially important point is that the introduced plasmids could be maintained in the organelles indefinitely, even though they could not detect the plasmids after 30-40 generations. Would it be possible to remove the origin of replication from the plasmids so they could not replicate in the organelles, and therefore disappear quickly from progenies?

There will be many challenges to editing of the organellar DNAs in other eukaryotes, given that there will be many mitochondria and/or chloroplasts in cells, requiring the introduction of a selectable marker such as antibiotic resistance. This will have regulatory issues over the long term. This point may need to be addressed in the manuscript. Nevertheless the technology is robust for basic research.

The evaluation of edited sites by restriction enzyme analyses (for example lines 446-451) is not very robust approach, it would be much better to sequence the regions and report any unanticipated sequence changes or rearrangements.

Overall, the observation that unintended indels and substitution events were extremely rare is a very positive indication that editing of the organellar DNAs will become an important tool!!

Experimental design

Robust and appropriate

Validity of the findings

Very convincing results

Additional comments

see above

---

## Round 0.2 · Major Revisions

At present, the author's manuscript still has some serious problems (as per Reviewer 1). The authors need to revise the article carefully.

Reviewer 1 ·

Basic reporting

Well written and understandable. Lt of unnecessary on line supplemental information.

Experimental design

Byung-Chun Yoo and colleagues report on DNA engineering in Chlamydomonas chloroplasts and yeast mitochondria using an independently replicating episomal element as an expression platform. The “Edit Plasmids” carry the Cas9 endonuclease gene, sgRNAs processed by tRNAs, the selectable plastid or mitochondrial marker, and a GFP flanked by short target sequences for insertion at the target site. Having short, homologous sequences on the donor DNA was a fatal flow in the experimental design. However short, the homology could direct integration by homologous recombination. To provide irrefutable evidence for CRISPR/Cas directed integration, authors should repeat the experiment without any targeting sequence on the donor DNA. They should then document that the integration of the cassette occurs at a different site, site dependent of the choice of sgRNA cutting the genome at a different location. NHEJ typically results in the loss or gain of at least a few nucleotides. Perfect homology at the target site is a red flag, the hallmark of homologous recombination.

Validity of the findings

see above.

Additional comments

Please, repeat experiment without targeting sequences on donor DNA.

Reviewer 2 ·

Basic reporting

Since the authors avoided clarifying the points raised in the comment #1 of my initial review, the actual applicability of the authors' method is still questionable. However, as the authors claimed, the phenomenon itself (i.e. CRISPR/Cas9-mediated organelle editing) was validated in this study regardless of its efficiency. Therefore, I basically accept the authors' claims.

Experimental design

See general comments.

Validity of the findings

See general comments.

Additional comments

1) According to the authors' response, "the goal presented in this manuscript was to test whether or not CRISPR/Cas9 works in organelles at all," and the editing efficiency is still unclear. This point should be clearly described in the main text. The actual applicability cannot be guaranteed without showing the data covering this reviewer's requests in the initial review.

2) The reference information should be added in line 662.

---

## Round 0.3 · Major Revisions

The author should obtain the data mentioned by Reviewer 1 as soon as possible.

Reviewer 1 ·

Basic reporting

The Topical Review by Thurtle-Schmidt and Lo entitled “Molecular Biology at the Cutting Edge: A Review on CRISPR/CAS9 Gene Editing for Undergraduates” cited by Corresponding Author is based on engineering the nuclear genome. Integration of foreign DNA in the nucleus in the absence of a double-strand break, by homologous recombination alone, is very rare, observed in 1 out of 100,000 integration events (or less frequent). If a double-strand DNA break is introduced, this frequency is boosted to 1/5. The role of double-strand DNA break in facilitating insertion of sequences at a specific location in the nuclear genome is well accepted. For discussion see for example Wright et al. (2005) High-frequency homologous recombination in plants mediated by zinc-finger nucleases. Plant Journal 44(4) 693-705.

In contrast, in chloroplasts and mitochondria there is no need for double-stranded breaks to observe homologous recombination, because homologous recombination is the rule. The question is what role, if any, double-strand DNA break plays when homology is provided for the insertion of heterologous DNA. This reviewer does not doubt that precise homologous recombination can be obtained using CRISPR/Cas9. This reviewer doubts whether or not the CRISPR/Cas9 construct reported here is active at all in either chloroplasts or mitochondria, because identical outcomes can be obtained in chloroplasts and mitochondria without CRISPR/Cas9, by homologous recombination.

The lack of evidence for non-homologous end joining (Analyses for on-target mutations ..) reported here could be the consequence having non-functional CRISPR/Cas9 vectors. Mutations inactivating chloroplast genes could provide a simple and convincing outcome regarding the functionality of the CRISPR/Cas9 vectors. Two chloroplast photosynthetic genes could be targeted in two different cultures and defective (photoheterotrophic) mutants collected from each. In cultures where the first gene is targeted, most phototrophic mutants should have mutations in the first gene at the predicted target site. If the second gene is targeted, the mutations should be in the second gene. An elegant, recent example for non-homologous end joining in mitochondria was provided by Kazama et al. in Nature Plants 5: 722-730, 2019, using TALEN-mediated gene inactivation. Authors could design similar experiments using their mitochondrial system.

Independent confirmation of CRISPR/Cas9 activity of the reported constructs should be the prerequisite of publication. The possibility of alternative explanation, integration by homologous recombination, should be pointed out.

Experimental design

see above

Validity of the findings

see above

Additional comments

see above

Reviewer 2 ·

Basic reporting

No additional comments

Experimental design

No additional comments

Validity of the findings

No additional comments

Additional comments

No additional comments

---

## Round 0.4 · Minor Revisions

Please answer the issues raised by the reviewers

Reviewer 1 ·

Basic reporting

Data supporting genome editing in yeast mitochondria is significantly strengthened in the revised manuscript. While in the previous manuscript only a single event was reported, in the current manuscript editing was detected in 15/15 samples when Cas9 is present, and 0/15 samples when Cas9 is absent. Because genome editing is reproducible in yeast mitochondria, the data are suitable for publication. However, manuscript should be improved to make it acceptable.

1) It would be helpful to explain in more detail what constitutes a sample in the results and document the data.

2) When preparing the manuscript, please, omit all data and reference to CRSPR/Cas9 mediated gene insertion in Chlamydomonas chloroplasts, because the chloroplast CRISPR/Cas9 constructs are not functional.

3). Authors correctly suggest that the chloroplast CERISPR/Cas9 construct is probably inactive when write: “The results indicate that the SNPs detected at the AvaII site in experimental lines and control lines are statistically similar and not a consequence of Cas9- induced SNPs. Enrichment of mutations at the AvaII recognition site is not due to Cas9 activity since they are found in similar frequency in the control lines without Cas9 or sgRNA. They likely represent enrichment of preexisting native polymorphisms or PCR errors at the AvaII site by our strategy for enriching the AvaII resistant clones. Based on our data, we did not find evidence for NHEJ DNA repair in chloroplasts.”

NHEJ is not the only mechanism of double-strand break repair. Double-strand DNA breaks in chloroplasts, when occur, frequently leave behind telltale signs, such as deletions repaired by microhomology-mediated end joining (seed Odom et al, cited). Because such deletions were absent in the present experiments, the simple explanation is that no double-strand breaks formed, because the CRISPR/Cas9 constructs is inactive.

4) Inactive CRISPR/Cas9 constructs explain the lack of statistical support for CRISPR/Cas9 mediated insertion in chloroplasts: 2 events in the YP13 pool of 20; and 0 events in the control pool of 32 clones (YP21 and YP23). The two events could be the products of rare homologous recombination.

Minor points:
5) In Figure 2, double underlining is not visible.
6) For information on linker design which yield fluorescent gfp fusion proteins see Zhou et al. (2011) A recombineering‐based gene tagging system for Arabidopsis. The Plant Journal 66, 712–723

Experimental design

Innovative.

Validity of the findings

Conclusions are acceptable in case of yeast mitochondria. Unacceptable in case of Chlamydomonas chloroplasts.

Additional comments

Manuscript should be revised to contain data only on yeast mitochondria.

Reviewer 2 ·

Basic reporting

See general comments.

Experimental design

See general comments.

Validity of the findings

See general comments.

Additional comments

The authors have adequately responded to the reviewer's comments.

---

## Round 0.5 · accepted · Accept

This is a very good manuscript supported by sufficient evidence.